# Melt focusing along lithosphere–asthenosphere boundary below Axial volcano

G. M. Kent[1,2 ✉], A. F. Arnulf[3,4], S. C. Singh[2], H. Carton[2], A. J. Harding[5] & S. Saustrup[3]

Beneath oceanic spreading centres, the lithosphere–asthenosphere boundary (LAB) acts as a permeability barrier that focuses the delivery of melt from deep within the mantle towards the spreading axis[1]. At intermediate-spreading to fast-spreading ridge crests, the multichannel seismic reflection technique has imaged a nearly flat, 1–2-km-wide axial magma lens (AML)[2] that defines the uppermost section of the LAB[3], but the nature of the LAB deeper into the crust has been more elusive, with some clues gained from tomographic images, providing only a diffuse view of a wider halo of lower-velocity material seated just beneath the AML[4]. Here we present 3D seismic reflection images of the LAB extending deep (5–6 km) into the crust beneath Axial volcano, located at the intersection of the Juan de Fuca Ridge and the Cobb–Eickelberg hotspot. The 3D shape of the LAB, which is coincident with a thermally controlled magma assimilation front, focuses hotspot-related and mid-ocean-spreading-centre-related magmatism towards the centre of the volcano, controlling both eruption and hydrothermal processes and the chemical composition of erupted lavas[5]. In this context, the LAB can be viewed as the upper surface of a 'magma domain', a volume within which melt bodies reside (replacing the concept of a single 'magma reservoir')[6]. Our discovery of a funnel-shaped, crustal LAB suggests that thermally controlled magma assimilation could be occurring along this surface at other volcanic systems, such as Iceland.

The LAB is fundamental to plate tectonics, with the overriding, brittle lithosphere 'floating' on a ductile, convecting asthenosphere, allowing for plate motion such as divergence at mid-ocean spreading centres[7] (for example, East Pacific Rise). The plethora of geophysical and geologic techniques used to examine this boundary has resulted in several complementary definitions. In a simplistic view, this boundary separates two distinct thermal regimes: conductive heat transport above the LAB versus convective heat transport below. The LAB can also be viewed as a boundary from a mechanical, rheological or compositional perspective[8]. Taken together, these definitions help to outline, beneath mid-ocean spreading centres, a tent-shaped boundary that focuses melt rising from deep within the mantle towards the crust, leading to eventual eruption at the seafloor or accretion of gabbroic rocks that form the mid to lower crust[9]. Seismic and electromagnetic investigation across the LAB reveals slower velocities and higher electrical conductivities, suggestive of a warmer and weaker regime beneath this boundary that is attributed to the presence of melt[10,11]. Also, geodynamic modelling suggests that melt extraction[12] provides a mechanism to pool melt at the base of the LAB that is consistent with seismic and electromagnetic observations near oceanic spreading centres[13,14], including beneath the Juan de Fuca plate east of Axial volcano[15]. The nature of this boundary in the crust and uppermost mantle in the vicinity of spreading centres, however, is uncertain, except for the presence of a narrow, 1–2-km-wide mid-crustal AML that has been observed at intermediate-spreading to fast-spreading centres and separates the brittle sheeted dykes and pillow basalts from the lower-crustal mush zone, including melt-rich sills stacked between the AML and Moho[16]. Thus, AMLs could be viewed as the shallowest, on-axis part of a broader LAB that deepens off-axis. Although tomographic images provide evidence of a wider halo of lower-velocity material seated just beneath the AML[4,17,18], extending 3–4 km on the ridge flanks, it has not been possible so far to image the LAB as a distinct boundary as observed for older lithosphere[1]. The ability to place geometrical constraints on the LAB deep into the crust beneath mid-ocean ridges would shed light on melt delivery into the lower to mid crust, which is critical to understand both eruptive and hydrothermal processes, and emplacement of the gabbroic section.

To investigate the dynamics of magma delivery, a 3D seismic reflection survey spanning 40 km × 16.3 km was collected aboard the RV Marcus Langseth in 2019 at Axial volcano, which sits at the intersection of the Juan de Fuca Ridge and Cobb–Eickelberg hotspot[19] (Fig. 1). Axial volcano hosts several hydrothermal fields[20] and has been the site of three eruptions[21–25] over the past 30 years. Axial volcano has a flat-topped summit that lies at a nearly constant water depth of about 1.4 km and hosts a horseshoe-shaped 8-km × 3-km caldera that separates north and south rift zones of the Axial segment along the Juan de Fuca Ridge. An earlier 2D seismic reflection survey was collected at Axial volcano in 2002 and imaged both a large 'main magma reservoir' (MMR) beneath the summit caldera and a 'secondary magma reservoir' (SMR)

[1]Nevada Seismological Laboratory, University of Nevada, Reno, Reno, NV, USA. [2]Université Paris Cité, Institut de Physique du Globe de Paris, CNRS, UMR 7154, Paris, France. [3]Jackson School of Geosciences, University of Texas Institute for Geophysics, Austin, TX, USA. [4]Amazon, San Diego, CA, USA. [5]Cecil H. and Ida M. Green Institute of Geophysics and Planetary Physics, Scripps Institution of Oceanography, University of California San Diego, La Jolla, CA, USA. ✉e-mail: gkent@unr.edu

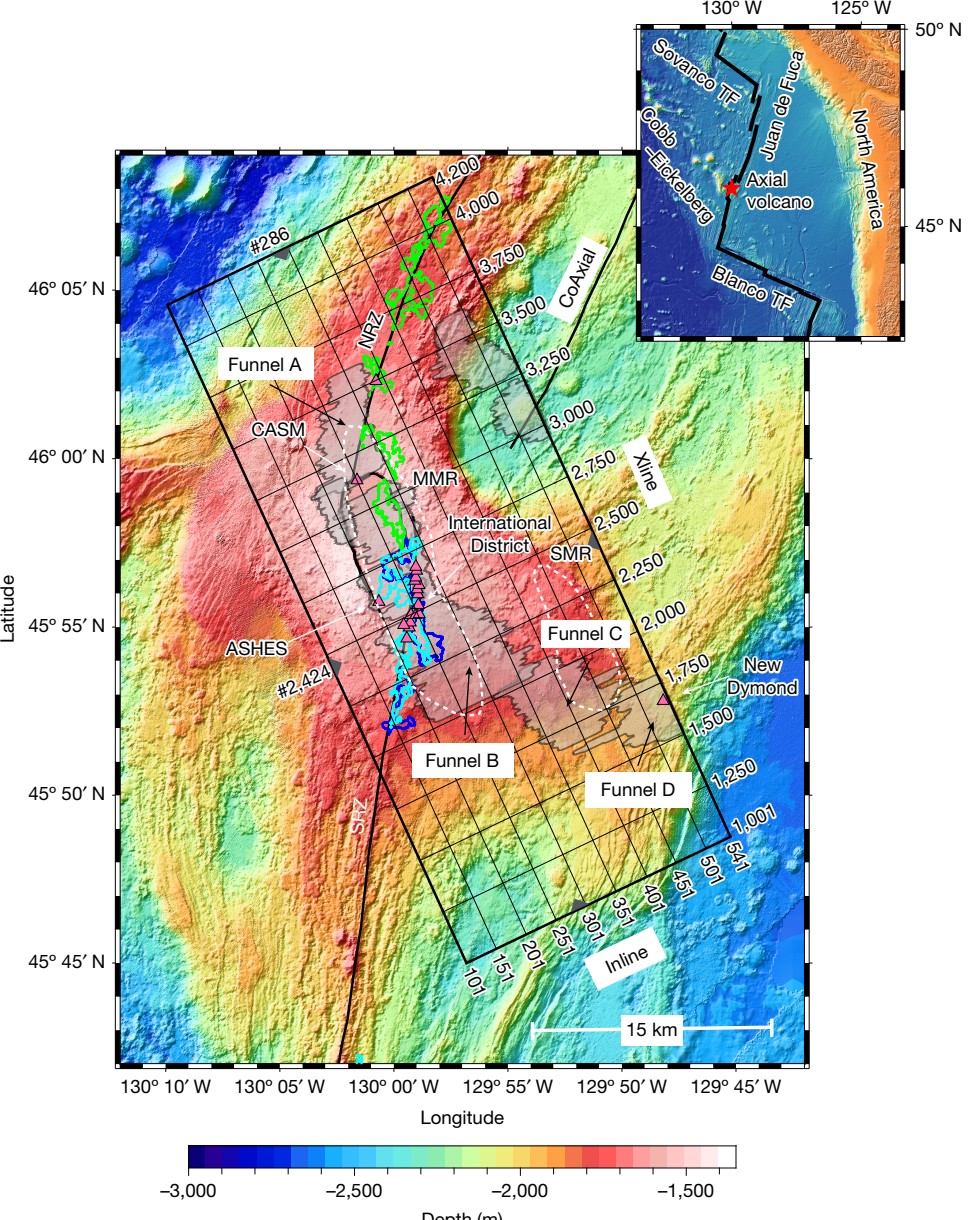

**Fig. 1 | Map of the Axial 3D experiment.** Bathymetric map of the Axial volcano region, including a 3D survey box (black rectangle) outlining the 40-km × 16.3-km region of seismic acquisition. Inset map provides a regional setting of the experiment within the Northeast Pacific. The 3D survey coordinate system is presented in an Inline (37.5-m spacing, perpendicular to direction of acquisition) and Xline (12.5-m spacing, parallel to direction of acquisition) reference frame. Both the north and south rift zones (NRZ and SRZ, respectively) and the adjacent CoAxial segment of the Juan de Fuca Ridge are shown as solid black lines. A solid black line also traces the edge of the horseshoe-shaped caldera wall. Grey triangles highlight the location of Inline profile #286 and Xline profile #2,424 (Fig. 2). AML|LAB funnel-shaped surfaces (transparent grey-shaded regions) are labelled (see black arrows) as funnels A, B, C and D and two more are shown on the CoAxial segment. Dashed white outlines of the previously identified MMR and SMR are shown[26]. Locations of hydrothermal fields CASM, ASHES, International District and New Dymond (small magenta-filled triangles; see white arrows) are shown[20]. Outlines of the 1998 (blue), 2011 (cyan) and 2015 (green) eruption lava flows are overlaid on the bathymetric map[59–61]. TF, transform fault.

southeast of the caldera[26]. These observations suggested the potential for complex interactions between adjacent magma reservoirs[26–30] but the dataset lacked sufficient density to clarify those interactions or link the reservoirs with the location of eruptive vents or hydrothermal fields. 3D seismic reflection data are markedly superior to 2D in terms of accuracy of the final images and provide better constraints on the spatial extent, continuity, orientation and dip of subsurface features, as demonstrated by previous 3D surveys along the East Pacific Rise centred at 9° 03′ N (ref. 31) and 9° 50′ N (ref. 32) and a similar survey at Lucky Strike volcano on the Mid-Atlantic Ridge at 37° N (ref. 33). Only through a comprehensive 3D seismic reflection survey at Axial volcano

is it possible to image the complex pattern of AML reflections, underlain in places by vertically stacked magma sills in the lower crust[16,27], to address the question of how and to what extent melt is focused into and through the crust and the relationships of these magma bodies with hydrothermal circulation and dyke initiation.

## 3D structure of the magma domain

This new 3D seismic reflection volume reveals the presence of large-scale, funnel-shaped structures extending deep into the lower crust (2.0 s two-way travel time or about 5–6 km below the seafloor (bsf);

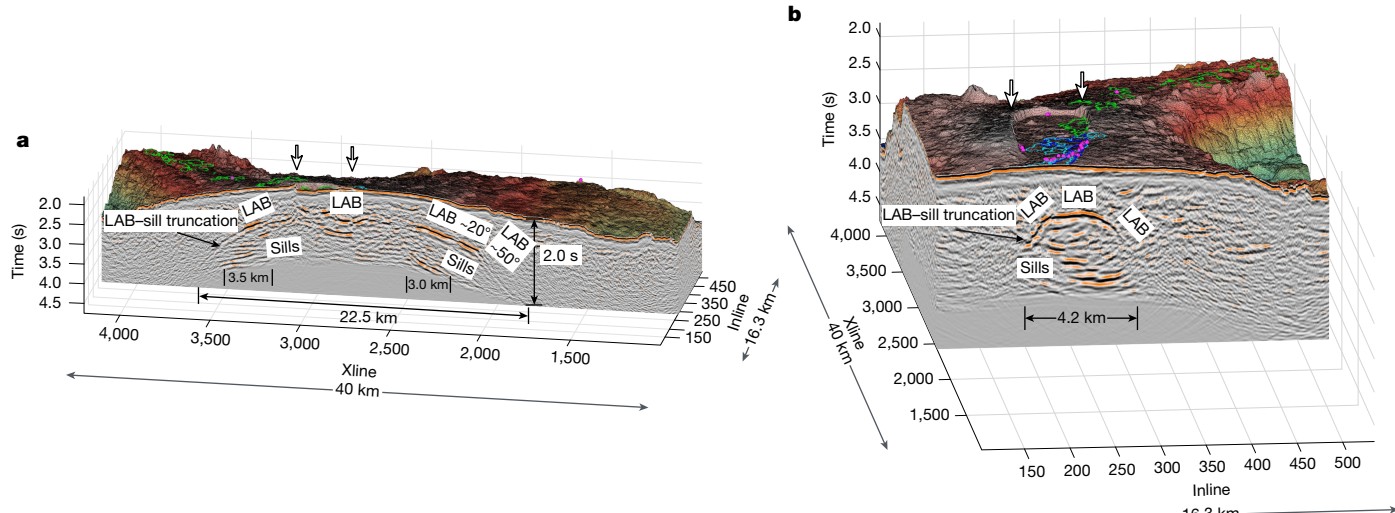

**Fig. 2 | Example of Inline and Xline reflection profiles.** 3D perspective view of Inline profile #286 (**a**) and Xline profile #2,424 (**b**) are shown beneath seafloor bathymetry. These profiles highlight the elongated-dome appearance of the AML|LAB beneath Axial volcano. The width of the LAB along the Inline slice is 22.5 km, whereas the Xline slice reveals a width of 4.2 km. Maximum depth of the AML|LAB (bsf) in two-way travel time of about 2.0 s is also indicated. In each perspective view, some of the magma sills beneath the AML|LAB are truncated (see black arrows) against this feature. Widths of magma sills are noted. Edges of the caldera are marked by white arrows. Locations of hydrothermal fields are shown as magenta-filled circles[20]. Outlines of the 1998 (blue), 2011 (cyan) and 2015 (green) lava flows are overlaid on the 3D bathymetric map[59–61]. Seismic data below the primary seafloor multiple have been greyed out. Inline (Supplementary Video 1) and Xline (Supplementary Video 2) animations provide a line-by-line viewing of the 3D seismic volume.

Fig. 2) that seem to be the crustal signature of the elusive LAB at young ages. These reflections arise from a seismic impedance contrast across a distinct boundary, resulting in the illumination of this widespread interface. With the ability to map the downward extensions of this 'expanded' AML deep into the lower crust (Figs. 1, 2 and 3 and Extended Data Figs. 1, 2 and 3), the distinction between the AML and the LAB begins to fade, as they are effectively one and the same structure (AML|LAB). Melt trapped along this interface, driven by buoyancy along a thermal boundary, attempts to migrate towards the apex of this structure, at which it is focused and primed for eruption or fails and solidifies to become the emplaced gabbroic section.

Notably, the AML|LAB, rather than being flat and sill-like, includes funnel shapes that do not mimic the seafloor topography beneath Axial volcano and its environs. These funnel-shaped features (Figs. 2 and 3 and Extended Data Fig. 1) constitute geometrical arrangements of melt that have never been seen before. They contrast with previously reported AMLs that are typically flat and whose depths are spreading-rate-dependent and controlled by the maximum depth of hydrothermal circulation based on both previous observations and theoretical studies[34–36]. These funnel-shaped structures are imaged at four locations beneath Axial volcano, of which two are northwest (funnel A) and southeast (funnel B) of the horseshoe-shaped caldera. These features are not aligned with the rift zones but are rotated counterclockwise approximately 25° and 35° to these features, respectively (Fig. 1). The combination of funnels A and B, plus reflectivity underlying parts of the caldera, represents a more detailed image of the earlier reported MMR[26]. Funnels A and B are connected beneath the central caldera (at a similar depth) through a series of smaller, disconnected magma sills, which are underlain by deeper, more continuous magma sills. A third funnel-shaped structure (funnel C) is also found beneath the extended bathymetric plateau of the volcano summit but lies east-southeast of the caldera (Fig. 3 and Extended Data Figs. 1 and 2). Last, a fourth funnel-like structure (funnel D) is seen beneath the New Dymond hydrothermal field and seems to overlap slightly with funnel C (Fig. 3 and Extended Data Figs. 1 and 2). The combination of funnels C and D provides a much-improved image and bounding shape of the SMR[26], whose original outline suffered from widely spaced 2D MCS

profiling and sparse ocean-bottom-seismometer-based ray coverage. Two of the four funnel-shaped structures (funnels A and B) are positioned to directly focus magmas into the caldera (Fig. 2), whereas the third (funnel C) and fourth (funnel D) structures terminate approximately 8 and 13 km east-southeast of the caldera edge, respectively (Figs. 1 and 3). The magmas underpinning funnels C and D are probably hotspot-sourced, as the footprint of the Cobb–Eickelberg hotspot is positioned in this region southeast of Axial volcano, as inferred from a slight increase in crustal thickness determined using wide-angle data[18], but there are no known recently erupted lavas above them. Other smaller magma reservoirs detected just outside the southeast opening of the caldera, imaged using full-waveform inversion techniques[30], are not well imaged in this post-stack migrated 3D reflectivity dataset.

Vertical slices perpendicular to the major axis of each of the funnel-shaped structures, or a combined transect along the apex of funnels A and B, clearly show an elongated-dome appearance to the AML|LAB, spanning about 5 km and 22.5 km, respectively, with dips ranging from near flat to 50° (Fig. 2). The morphology of the AML|LAB interface refines the idea of vertically stacked magma sills reported using conventional 2D profiles[16,27]. We find that the funnel-shaped AML|LAB bounds these underlying sills and, in many cases, truncates them (Fig. 2 and Extended Data Figs. 2 and 3). The fact that some of the underlying sills abut this reflector provides strong evidence that the bounding structure is the thermally controlled LAB. The lateral dimensions of these stacked sills are at most 3.5 km and, in some cases, much smaller. The shallowest section of the AML|LAB is located at the southeast edge of the caldera and is expressed as a narrow ribbon of magma that shoals in two separate areas separated by 2.1 km, both of which sit at the updip termination of funnel B (Fig. 4). These locations are also coincident with the highest concentration of hydrothermal vents that comprise the International District hydrothermal field[20] and the highest density of seismicity[37,38] and 'mixed-frequency' earthquakes[39]. Furthermore, this ribbon-shaped melt body seems to underlie the initiation point of all three documented eruptions in 1998, 2011 and 2015 (refs. 21–24).

The strength of the AML|LAB reflection depends primarily on the impedance (velocity × density) contrast across this boundary, although tuning within the AML can also cause variations in reflector strength[40].

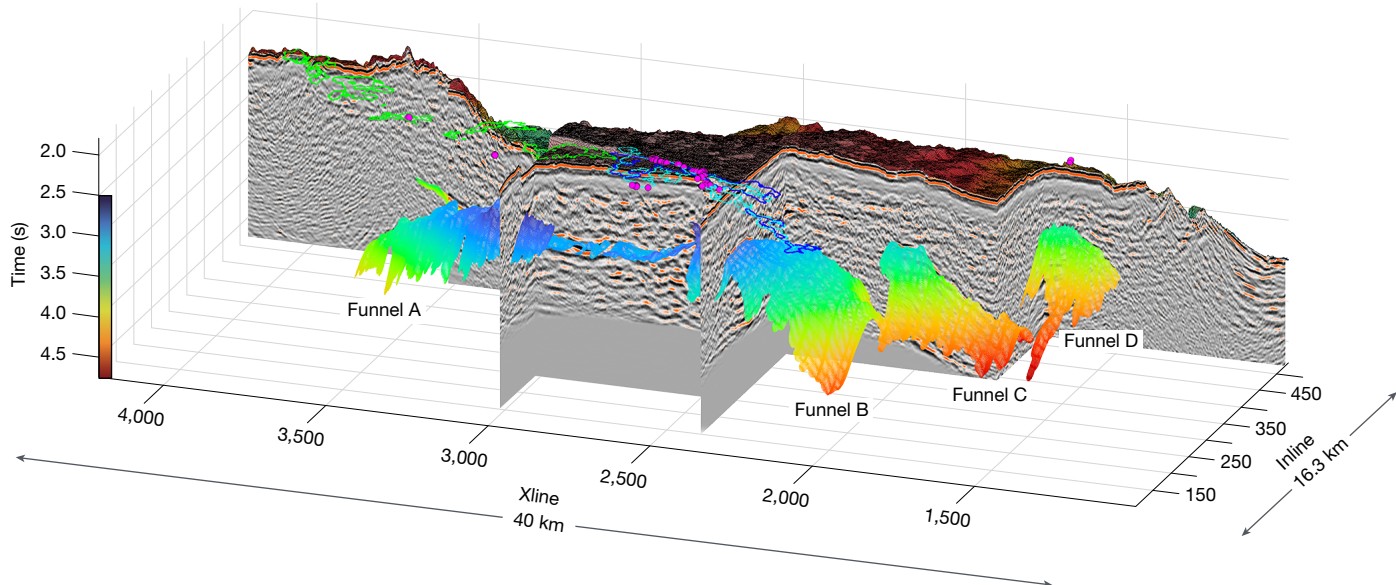

**Fig. 3 | Morphology of the AML|LAB boundary.** 3D perspective view of the funnel-shaped AML|LAB boundaries beneath Axial volcano. A portion of the 3D seismic volume has been removed to highlight the structure of this boundary relative to the remaining seismic data. A view towards the northeast reveals the AML|LAB structure of funnel A (northwest of the caldera), funnel B (southeast of the caldera), funnel C (far southeast) and funnel D (beneath the New Dymond hydrothermal field). The structure of the AML|LAB beneath the caldera is complex. The apexes of funnels B and C are offset 3.5 km from each other.

Funnels C and D also seem to be similarly offset (about 5 km). Semi-transparent AML|LAB surfaces of funnels A, B, C and D have been shaded by two-way travel time, in which cool colours represent shallower parts of the structure and warmer colours show the deepest areas; vertical colour bar (two-way travel time) is included on the left. Locations of hydrothermal fields are shown as magenta-filled circles[20]. Outlines of the 1998 (blue), 2011 (cyan) and 2015 (green) lava flows are overlaid on the 3D bathymetric map[59–61]. Supplementary Video 3 provides a full 360° tour of the shape and geometry of the AML|LAB structures.

Other contributing factors to amplitude variation include attenuation and topographic focusing effects. To explore a map of reflection strength (a first-order filter to separate regions of high melt concentration), we have used a voxel visualization strategy in which each seismic amplitude sample is represented as an elemental parallelepiped (a voxel in 3D being equivalent to a pixel in 2D) and only those data points with the highest seismic amplitude values are visualized[31]. In this way, regions beneath Axial volcano with the strongest reflections, and by proxy melt availability, can be viewed in three dimensions. Projected in map view, the strongest reflections underlie the hydrothermal fields that are located along the southeast edge of the caldera (Fig. 4). Strong reflections are also seen near the intersection between the north rift zone and the caldera, with the largest footprint, however, observed southeast of the caldera beneath the extended plateau (Fig. 4). Strong reflections are also coincident with the perched AML farther to the southeast. By contrast, the distribution of strong reflections directly beneath the caldera is spotty and less continuous relative to other regions already highlighted. The two shallowest reaches of the AML|LAB (0.56 s and 0.65 s two-way travel time or at approximately 1.1 km bsf minimum depth[26,28]), separated by 2.1 km, can be observed using voxel visualization (Fig. 4), which—as previously mentioned—may be near the initiation point for the recent eruptions. This narrow ribbon of high melt fraction is coincident with PmeltS reflections (a wave impinging on the AML as a P-wave, at which it gets converted to an S-wave, then converted back to a P-wave at the seafloor) that were observed from a 2D profile collected in 2002 (ref. 28); strong PmeltS reflections are observed where the AML consists of nearly pure melt[40].

Analogue studies related to inflation–deflation processes within volcanic systems[41] would predict that the shallowest reaches of this magma domain[6] would be the most favourable for dyke initiation and emplacement, consistent with the recent eruption history at Axial volcano. In fact, the southward propagation of dykes during the 1998 and 2011 eruptions and northward propagation in 2015 seem to be associated with the two peaks on the ribbon-shaped melt body

(Fig. 4), indicating that the shallowest anatomy of the magma bodies may control the eruption initiation and the direction of dyke propagation. Notably, dyke propagations occur along the Axial rift zones, not where the most expansive melt-rich bodies are observed in the subsurface, indicating that dyke propagations, and hence eruptions, occur in regions in which extensional stress is maximum (Fig. 4). The limited observation of strong AML|LAB reflections beneath the north and south rift zones[29] is consistent with the above observations. The other mismatch is between the area of maximum geodetic uplift in the years preceding eruptions (near the centre of the caldera)[42] and maps highlighting magma availability (Fig. 4). This disconnect may owe its existence to a system of 'weak' ring faults encircling the caldera[37] that, in turn, would amplify upward deformation of the caldera floor as the underlying magmas coalesce, in contrast to magmas pooling southeast of the caldera, at which a similar mechanism does not exist.

## Melt focusing and assimilation

Taking these new observations together, the simplest explanation for the large-scale funnel-shaped structures observed is that they represent a magma assimilation front, along which rocks from the overlying carapace are being remelted (Fig. 5). This is especially obvious at funnels C and D, at which magma has melted through the overlying lithosphere, resulting in perched magma bodies that are located a considerable distance from either the volcano summit or its associated rift zones (Extended Data Fig. 2). Therefore, the AML|LAB imaged here probably represents an assimilation front controlled by the thermal structure at Axial volcano, in a configuration such as that of melt underpinning the LAB for older oceanic lithosphere[1].

Southeast of the caldera, the downdip termination of funnel B is coincident but laterally offset 3.5 km from the updip termination of funnel C (Fig. 3). If viewed from a perpendicular slice through the major axes of funnels B and C at this offset, the dome-shaped AML|LAB structures 'connect' at depth, suggesting that magmas beneath funnel C

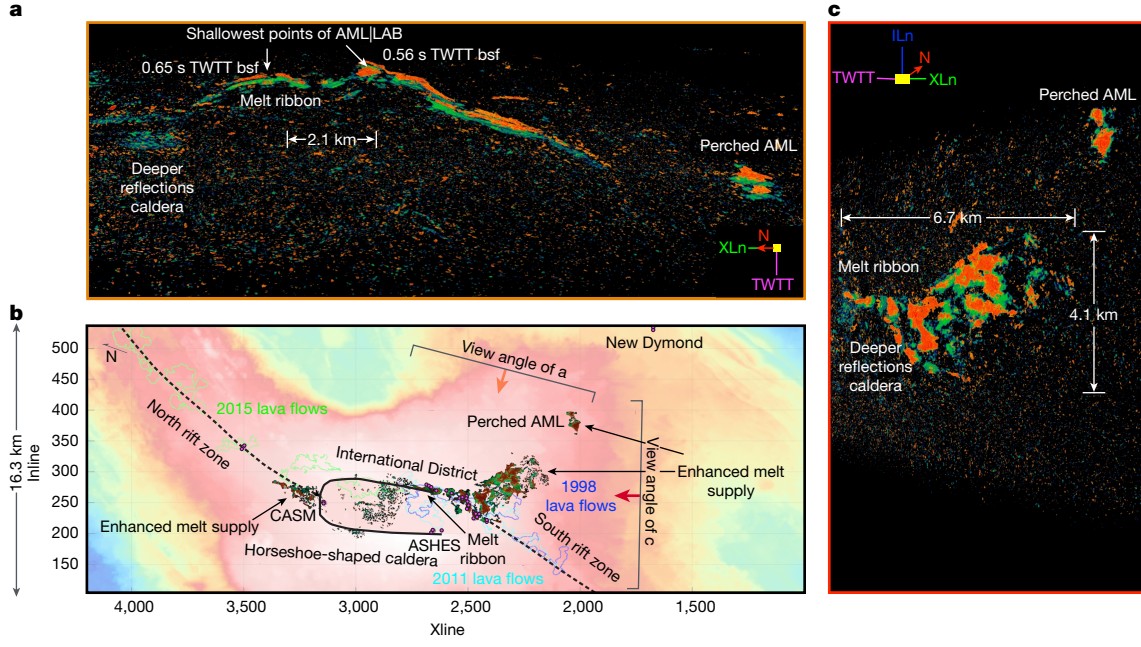

**Fig. 4 | Melt availability map beneath Axial volcano.** 3D perspective of strong seismic amplitudes from side-on perspective (**a**), top-down perspective (**b**) and oblique, overhead view (**c**). **a**, A perspective view looking edge-on into the reflectivity volume towards the southwest, highlighting strong reflections beneath Axial volcano. Two shallow points of the AML|LAB structure are clearly visible near the northeastern caldera wall. The view angle and image bracket (orange arrow in **b**) provides the orientation of the image. **b**, Seismic amplitudes are projected on top of the bathymetric data within the 3D seismic footprint. A solid black line traces the edge of the horseshoe-shaped caldera wall and dashed lines show the south and north rift zones. **c**, An oblique, overhead 3D view from an observation point southeast of Axial volcano, highlighting the widespread presence of strong seismic amplitudes beneath the southeast flank of the volcano. The view angle and image bracket for this view is shown by the red arrow in **b**. A narrow melt ribbon along the northeast wall of the caldera contrasts with widespread strong reflectivity across the southeast flank of the volcano. The 3D volume has been divided into voxels, for which only those voxels with the highest amplitudes (either positive or negative), coloured red-orange and green-blue, have been rendered between 2.628 s and 3.3824 s two-way travel time (TWTT). A compass (in **a** and **c**) with Xline (XLn), Inline (ILn), TWTT and North (N) orientations is also provided for orientation. Locations of hydrothermal fields CASM, ASHES, International District and New Dymond (small, magenta-filled circles) are highlighted[20]. Outlines of the 1998 (blue), 2011 (cyan) and 2015 (green) lava flows are overlaid (**b**) on the 3D bathymetric map[59–61]. Supplementary Video 4 provides a 3D tour of the voxel visualization, highlighting areas of strong reflectivity beneath Axial volcano.

could 'leak' into funnel B along a subsurface fracture or failure plane. The downdip extension of funnel B probably exists but is difficult to image owing to a lack of sufficient melt trapped beneath this boundary at present. The ability to detect this boundary, which is related to the accumulation of magmas beneath it, will be greater during an episode of increasing magma supply from the mantle; such an episode in turn drives an active phase of assimilation as magma melts through the overlying brittle rocks.

A short-wavelength, shoaling AML morphology is observed at a much smaller scale at three locations along the southern and northern East Pacific Rise at 17° 20′ S (refs. 34,43) and 8° 52′ N (ref. 44) and the Cleft segment of the Juan de Fuca Ridge at 44° 46′ N (ref. 16), respectively, at which the overlying sheeted-dyke complex seems to be thinned by at least roughly 400 m over a lateral distance of about 5 km; poor seismic coverage at these locations, however, provides uncertainty to the origin and extent of these anomalies. Extended or wide AML reflections are also seen just north of the 9° 03′ N overlapping spreading centre on the East Pacific Rise[31,45]. A decoupling in melt supply between the northern and southern limbs of the overlapping spreading centre may allow magmas to accumulate beneath the LAB, producing anomalous AMLs that are imaged at about 4 km in cross-axis width. A similar geometry may help explain the AML|LAB imaged near the CoAxial segment within this dataset (Fig. 1), at which an extended AML|LAB reflection is imaged pointing updip towards the northern rift zone of Axial volcano.

As the AML|LAB boundary plunges deep into the lower crust, active phases of magma assimilation would necessarily involve remelting of both the sheeted dykes (above the AML) and gabbroic rocks (above

the downdip AML 'extensions') back into the reservoir (Fig. 5). The existence of magma assimilation at the gabbro−dyke boundary has been well documented[5,46], but there have been limited observations of the assimilation of gabbroic rocks in the lower crust[47]. At IODP hole 1256D, the lava sequence is 650 m thick, whereas the foreshortened dyke sequence is only 350 m thick[46], which could be because of the assimilation of the sheeted dykes at this location. Near the caldera at Axial volcano, the average velocity structure of the upper crust is characterized by an approximately 600−800-m-thick layer 2A (lava flows), underpinned by localized shoaling of the AML|LAB depth to about 1,100 m bsf (ref. 26), which suggests a thinned layer 2B (sheeted dykes) of about 300−500 m thickness. This observation mimics what was observed at IODP hole 1256D, providing further evidence for substantial assimilation along this interface beneath Axial volcano. The estimated dips of the AML|LAB interface at Axial volcano range from nearly flat up to 50°, which mimics the measurements of magmatic layering and foliation seen in the upper gabbroic sequence within the Oman ophiolite with dips that range from 0° to about 40°, giving an average dip of 18−19° (ref. 48). Melt migration and assimilation along the AML|LAB interface may provide a mechanism to impart these observed dips that are subsequently locked in as the thermal boundary migrates and freezes owing to plate spreading.

As assimilated rocks contain hydrous minerals formed owing to the deep-reaching hydrothermal circulation, associated reservoir contamination would cause changes in the chemistries of erupted lavas towards more plagioclase-rich trends[49,50] and, to an extreme, silica-rich magmas are possible, as observed in Iceland[51,52]. The evolved lavas observed at

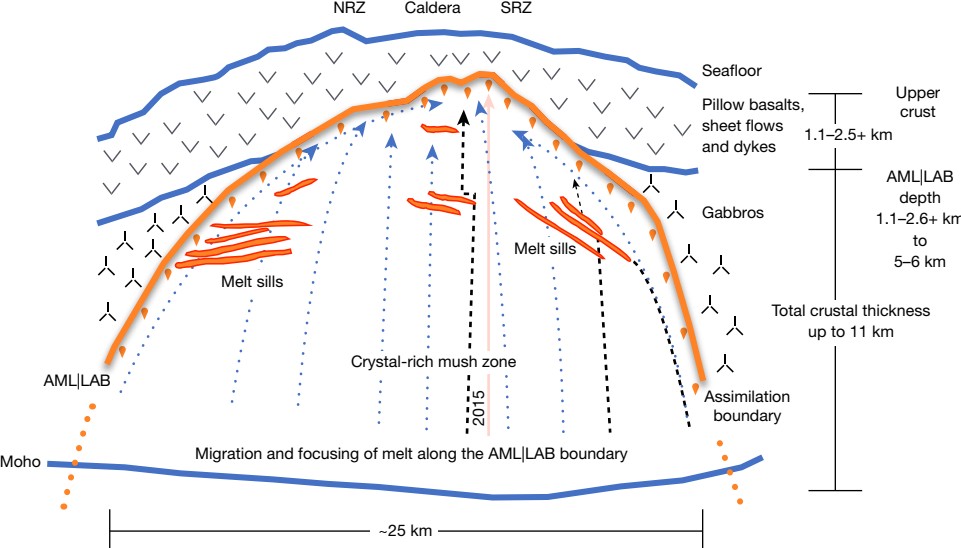

**Fig. 5 | Melt focusing and assimilation along the AML|LAB boundary.**
Schematic highlighting the focusing of melt along the AML|LAB towards the caldera and assimilation of upper-crustal and gabbroic rocks across this boundary. Solidified upper crust and gabbroic rocks are shown, with interfaces between these layers drawn as blue lines, including the base of the crust (Moho). Potential assimilated rocks underlying the present AML|LAB are shown by inverted drops (orange). The interpreted AML|LAB assimilation boundary is shown as an orange line; discontinuous regions along the AML|LAB boundary with strong reflections (high melt fraction) are shown in red. Melt sills beneath the AML|LAB are also shown in orange traced by red edges. Idealized melt transport is indicated by near-vertical ascent arrows (with blue dots) until the AML|LAB boundary is encountered, after which melt migrates along this interface. Some melt ascending through the crystal mush either vertically or along the AML|LAB (arrows with dashed blacked lines) may encounter sub-AML|LAB melt sills, at which melt may spend time in isolation before potentially ascending. A large semi-transparent red arrow (centre) suggests possible melt bypass in 2015, during which more primitive, probably deep-seated, melts were erupted. Approximate thickness[26] of oceanic layers, depth to the AML|LAB[26] and maximum crustal thickness[18] are shown (on the right). NRZ, north rift zone; SRZ, south rift zone.

Axial volcano[53] and at many mid-ocean spreading centres[5] were thought to involve petrologic differentiation through isolation, as potentially provided by sub-AML|LAB sills[54]. Our new findings, however, would suggest that two mechanisms simultaneously exist beneath Axial volcano (Fig. 5): (1) stacked vertical sill structure may provide pathways that promote isolation of magmas, giving rise to greater differentiation, and (2) through assimilation of the overlying hydrated substrate, an independent mechanism exists to drive differentiation towards more evolved magmas. At Axial volcano, these two mechanisms exist together and could potentially work in tandem to enforce trends towards further differentiation to help explain the petrologically differentiated lavas (Group 1 with low Mg # <7.9) observed at this site[53]. The more mafic lavas such as those observed during the first part of the 2015 eruption (Group 2, high Mg # >7.9) are probably not sourced from magma that resides along the AML|LAB interface but rather from a deeper source that mostly bypasses both contaminated melts and underlying sills and is directly emplaced into the dyke feeder system[23,55]. Petrological studies of historical Axial lava flows predominantly suggest an evolved or Group 1 origin; by contrast, the last eruption in 2015, and a time span from 1200 to 1400 CE, contained less evolved Group 2 lavas. Together, these observations suggest that Axial volcano has been constructed more often by evolved lavas that have migrated along the AML|LAB boundary and/or were isolated in sub-AML|LAB sills, with less frequent contributions by primitive melts that bypass mid-crustal to lower-crustal processes and directly erupt onto the seafloor (Fig. 5). A subset of 2015 lavas along the northern rift zone that are more petrologically evolved[23,55] may have tapped into the contaminated melt zone along the AML|LAB interface near the intersection of the northern rift zone and the caldera (Fig. 4).

## Temporal variability of the LAB

The ability to image the widespread AML|LAB interface seems to be evidence that a strong magmatic phase is now underway at Axial volcano[23,53,55]. The lack of notable seafloor topography relative to most mid-ocean spreading centre environments also provides near-ideal conditions for seismic imaging at depth. These factors have helped enable a new view into the dynamics of melt delivery near the base of the crust and redistribution along the AML|LAB towards the caldera. These images also lend support to a mechanism in which hydrated rocks are assimilated, which can impart substantial influences on the chemistry of lavas erupted at Axial volcano and, more generally, elsewhere. When the present phase of robust magmatism finally wanes at Axial volcano, so will the degree of assimilation, and solidification/accretion along this boundary will take place as the size and extent of the AML|LAB reduces accordingly. Within this future magma-restricted environment, one might expect less petrologic diversity as both contamination through assimilation and multisill-based isolation becomes less common and so does the prevalence of evolved lavas. In turn, the absence of a robust magma supply will make imaging the downdip extensions of this boundary more difficult, resulting in a shrinking AML|LAB beneath Axial volcano that becomes indistinguishable from smaller AMLs seen beneath other spreading centres. Thus, these episodic upswings or supercycles of magma delivery into the crust not only help to reveal the downdip extent of the LAB but also leave, within erupted lavas, a cyclical record of contamination through assimilation of erupted lavas[50] that may explain the bimodal distribution of lavas erupted at Axial volcano[53].

Our results have implications for other large volcanic provinces formed by ridge–plume interaction, such as Iceland. In Iceland, a thick crust (30–40 km)[56] is formed by the interaction between the Iceland plume and the Mid-Atlantic Ridge. As it is difficult and expensive to seismically image magmatic systems on land, the nature of the crustal AML|LAB beneath Iceland is poorly understood. Nevertheless, a record of extensive assimilation has been observed in the petrology of erupted lavas[51,52]. We suggest that sills in Iceland are probably distributed more widely in three dimensions and are probably present near

the AML|LAB boundary, at which assimilation occurs. A similar process may also occur at slow-spreading and ultraslow-spreading ridges, at which the AML|LAB boundary is probably 3D and ephemeral and where melt would migrate along the AML|LAB boundary towards the segment centre, forming a central volcano, such as at the Lucky Strike segment[33] in the Atlantic and the 50° 28′ E segment on the Southwest Indian Ridge[57]. The petrology of lavas from the above volcanic centres is consistent with assimilation[50,58], implying that a crustal AML|LAB and magma assimilation might be prevalent in other mid-ocean-ridge settings.

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

## Methods

### Data acquisition, navigation and binning

The Axial 3D acquisition geometry used aboard the RV Marcus Langseth included four 5.85-km-long Sercel solid-state hydrophone streamers staggered at 150-m spacing. Two flip-flopping, 54.1-l tuned-airgun arrays were towed 75 m apart; both streamers and source arrays were deployed symmetric to the ship centreline. The source was towed at 12 m and the four streamers at 16 m water depth, providing a low-frequency energy source to penetrate deeper into the subsurface. Navigation of the four 468-channel streamers, with hydrophones spaced 12.5 m apart, were facilitated by Digicourse compass birds, Sonardyne acoustic net modules and four Western Geophysical GPS-enabled tail buoys. Near-real-time streamer positioning was facilitated using Spectra, Sprint and Reflex modules for navigation capture, QC and binning analysis. This swath 3D geometry provided eight parallel common midpoint (CMP) lines separated by 37.5 m in the crossline direction; the inline binning of the seismic data was set at 12.5 m or twice the natural spacing of 6.25 m between CMP lines, yielding a nominal fold of 78. Flexible binning was not used in the seismic data volume presented here. The 40-km × 16.3-km acquisition box was constructed from 53 sail-line passes, ten further inline passes to ameliorate the most serious gaps in data coverage and six further lines were reshot.

### Stacking and migration

Careful pre-processing of the 3D seismic data included optimal computation of two predictive deconvolution filters for each sail line (one filter for each of port and starboard sources) to minimize ringing effects from the airgun source. Swell noise attenuation was facilitated using an expectation-maximization algorithm[62]. Trace interpolation was performed to infill all dead channels, whereas optimal trace balancing of seismic data was implemented for each sail line to ensure the continuity of the seismic wavefield. Data were also filtered using a Butterworth bandpass filter with corner frequencies of 2–45 Hz. A surgical mute was applied to CMP gathers to remove refracted energy. The CMP gathers were corrected for normal moveout and stacked (making use of offsets up to about 2.5 km for the shallow AML event) using r.m.s. velocities derived from an existing high-resolution tomographic inversion[26]. A 3D finite-difference $f$-$x$ post-stack time migration scheme was performed using interval velocities[26] to produce geometrically correct images.

### Data availability

MCS data for the 3D cruise are available on the Marine Geoscience Data System website at https://www.marine-geo.org/; the cruise information name is MGL1905, https://doi.org/10.7284/908292. The bathymetric and topographic data used in this study are available at GEBCO – the General Bathymetric Chart of the Oceans (https://www.gebco.net).

62. Bekara, M. & van der Baan, M. High-amplitude noise detection by the expectation-maximization algorithm with application to swell-noise attenuation. *Geophysics* **75**, V39–V49 (2010).

**Acknowledgements** We thank the officers and crew of the RV Marcus Langseth and supporting Office of Marine Operations at Lamont-Doherty Earth Observatory, Columbia University. We also thank the around-the-clock shipboard open-participation crew and onshore liaison: A. Cap, B. Oiler, M. Bellucci, M. Griffiths, M. Lee, M. Goulain, S. Brandt, T. Eischen, V. Lucas, S. Mitchell and A. Kell. We thank J. Beeson for assistance with the RV Marcus Langseth-derived multibeam bathymetry. Discussions with L. France on petrologic contamination were also extremely beneficial. We also thank B. Savran, D. Trugman and W. Wilcock for their thoughtful comments. Data-processing software was provided by AspenTech, Inc. (Paradigm 22). Echos was used for the 3D binning, stacking and migration of the multichannel seismic data. Data presentation of the 3D seismic volume was facilitated through visualization using 3D Canvas and MATLAB. Generic Mapping Tools (GMT) and MATLAB were also used for figure preparation. This project was financed through the National Science Foundation awards OCE-1658021 (UNR), OCE-1658199 (UTIG) and OCE-1658018 (UCSD). The idea of crustal lithosphere–asthenosphere boundary developed during the preparation of ERC Advanced Grant 101141564 MohoLAB. The IPGP contribution number is 4294.

**Author contributions** A.F.A., S.S. and A.J.H. collected the 3D dataset aboard the RV Marcus Langseth; the 3D dataset was processed by A.F.A., S.S. and A.J.H.; interpretation of the 3D dataset was undertaken by G.M.K., A.F.A., S.C.S. and H.C.; the manuscript was written by G.M.K., A.F.A., S.C.S., H.C. and A.J.H.

**Competing interests** The authors declare no competing interests.

**Additional information**
**Correspondence and requests for materials** should be addressed to G. M. Kent.

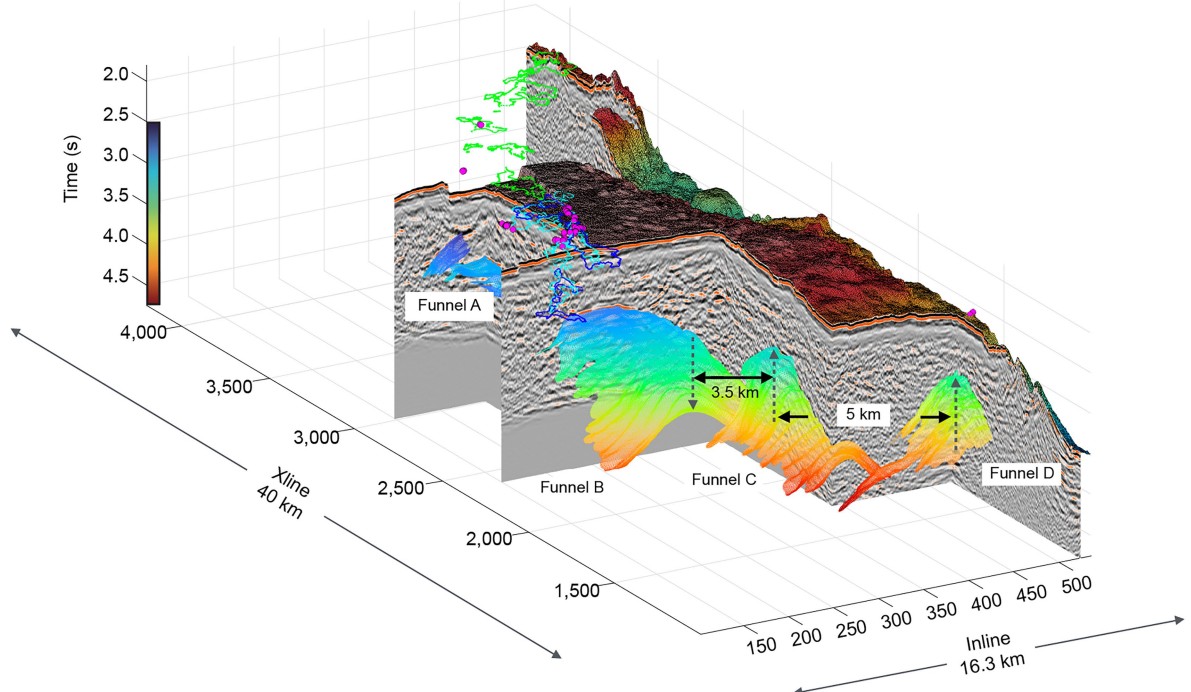

**Extended Data Fig. 1 | A different perspective of the AML|LAB boundary.** A different perspective of the AML|LAB with a view towards the north, which highlights the offset of the imaged AML|LAB between funnels B and C, with the latter located about 8 km from the caldera. The apexes of funnels B and C are offset 3.5 km from each other. Funnels C and D also seem to be similarly offset by roughly 5 km. Semi-transparent AML|LAB surfaces of funnels A, B, C and D have been shaded by two-way travel time, for which cool colours represent shallower parts of the structure and warmer colours show the deepest areas; vertical colour bar (two-way travel time) is shown on the left. Locations of hydrothermal fields are shown as magenta-filled circles[20]. Outlines of the 1998 (blue), 2011 (cyan) and 2015 (green) lava flows are overlaid on the 3D bathymetric map[59–61]. Supplementary Video 3 provides a full 360° tour of the shape and geometry of the AML|LAB structures.

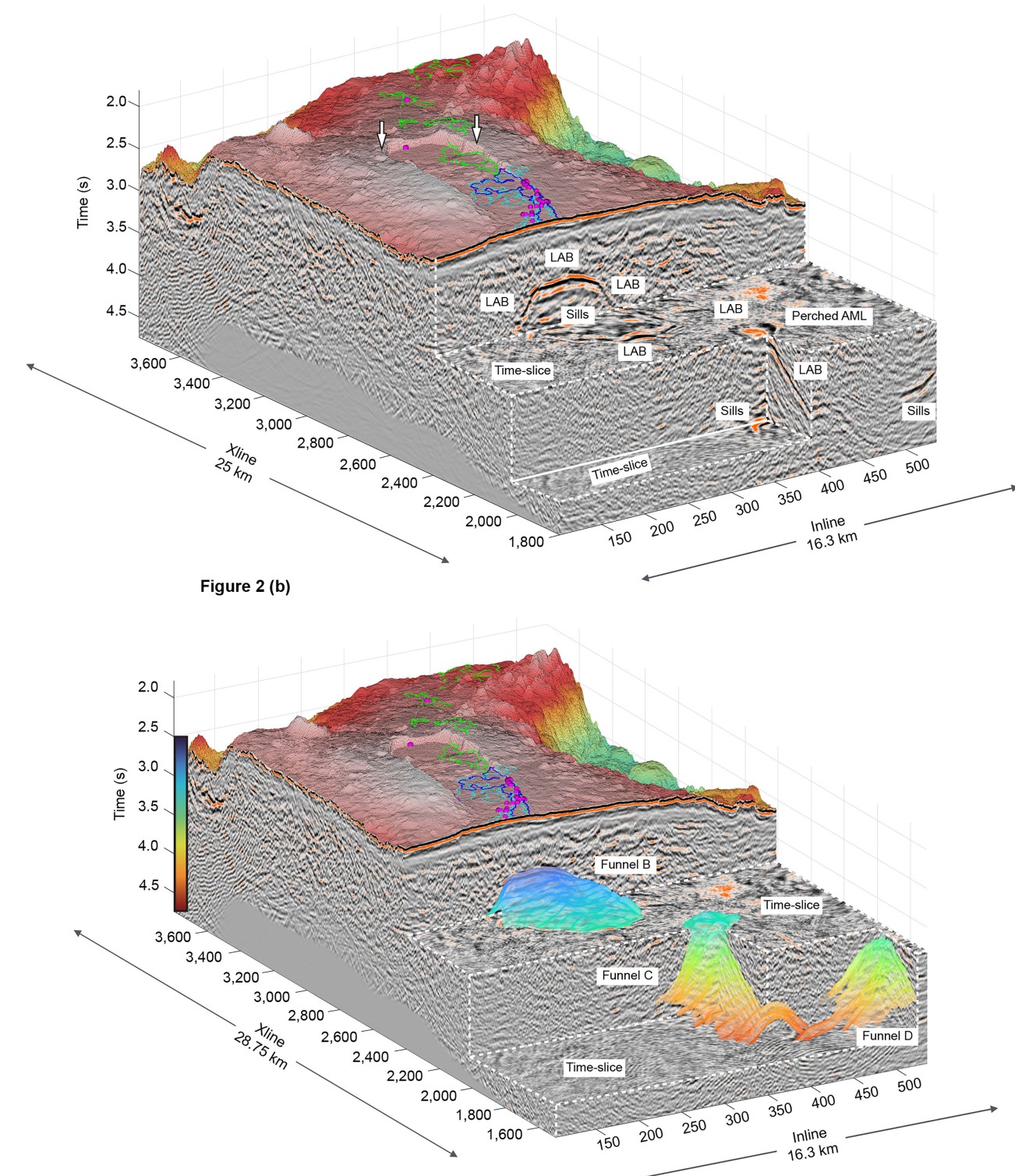

**Figure 2 (b)**

**Extended Data Fig. 2 | Seismic reflectivity and AML|LAB morphology southeast of caldera.** A sculpted 3D perspective of the AML|LAB structure beneath the southeast slope of Axial volcano, highlighting the relationship between two distinct AML|LAB structures (funnels B and C). **a**, The embedded 'chair-cut' visualization shows the top of the perched AML|LAB structure (see bullseye feature on the upper time slice). The primary front-facing slice is at Xline #2,411. The shape of funnel B can also be traced on the same time slice. Dashed white lines outline the two rectangular chair-cuts. The lower chair-cut allows the viewer to see stacked sills beneath the perched AML|LAB. **b**, Similar perspective for which semi-transparent AML|LAB surfaces of funnels B, C and D have been added and are shaded by two-way travel time, with cool colours representing shallower parts of the structure and warmer colours highlight the deepest areas; vertical colour bar (two-way travel time) is shown on the left. Edges of the caldera are noted by arrows. Locations of hydrothermal fields are shown as magenta-filled circles[20]. Outlines of the 1998 (blue), 2011 (cyan) and 2015 (green) lava flows are overlaid on the 3D bathymetric map[59-61]. Seismic data below the primary seafloor multiple have been greyed out. Supplementary Video 5 provides a tour of the shape and geometry of the AML|LAB structures beneath the southeast flank of Axial volcano.

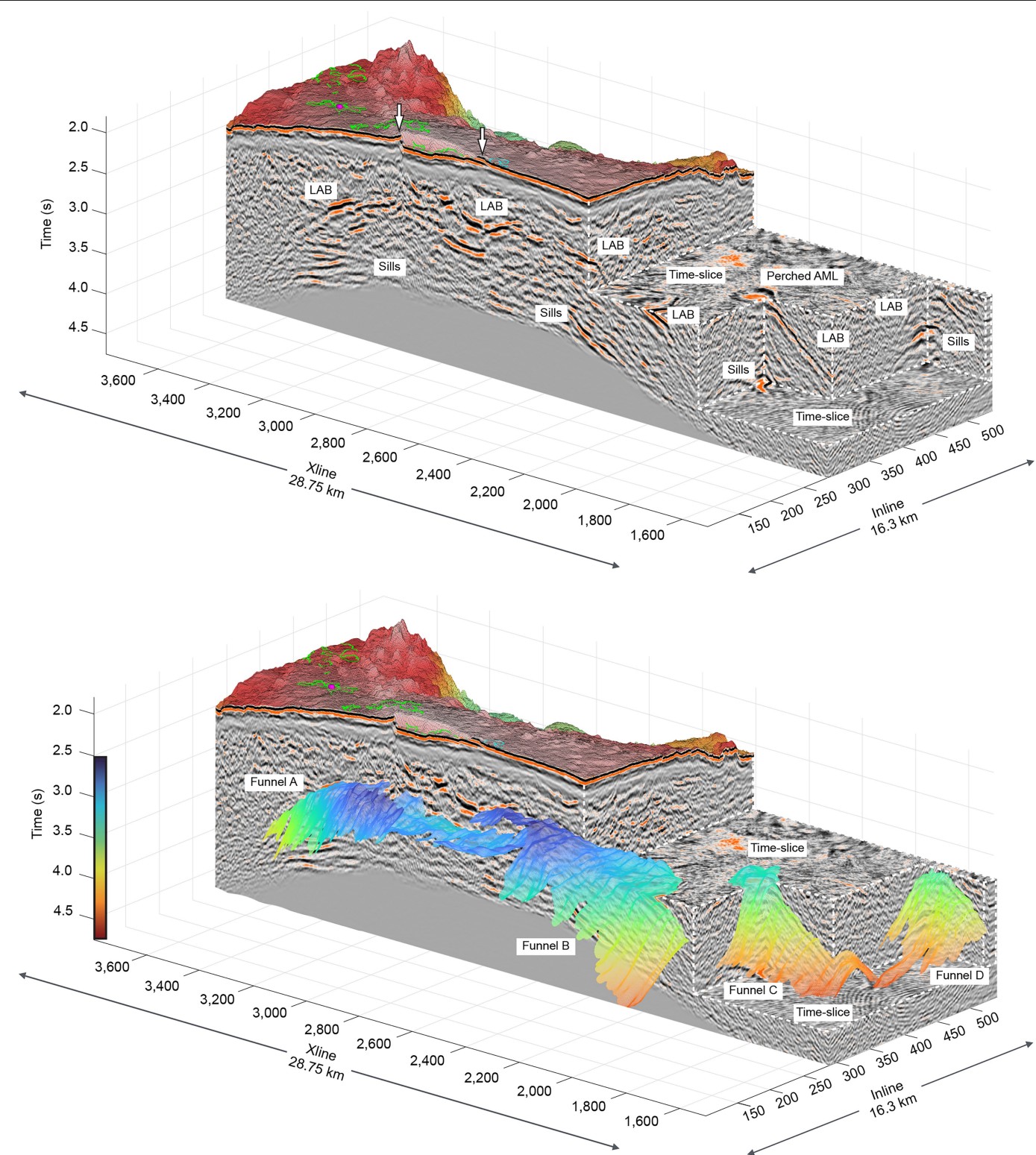

**Extended Data Fig. 3 | Seismic reflectivity and AML|LAB morphology beneath the caldera. a**, A complex, 3D cut-out view of the seismic reflectivity volume highlighting the structure of the AML|LAB across the four main magma bodies (not including the magma bodies located near the CoAxial segment). The AML|LAB and underlying sills are labelled. **b**, Similar perspective for which semi-transparent AML|LAB surfaces of funnels A, B, C and D have been added and are shaded by two-way travel time, with cool colours representing shallower parts of the structure and warm colours show the deepest areas; vertical colour bar (two-way travel time) is shown on the left. Edges of the caldera are noted by arrows. Locations of hydrothermal fields are shown as magenta-filled circles[20]. Outlines of the 2011 (cyan) and 2015 (green) lava flows are overlaid on the 3D bathymetric map[60,61]. Seismic data below the primary seafloor multiple have been greyed out. Dashed white lines outline the complex rectangular chair-cuts.