## [Peer review file · Nature]

Melt focusing along the lithosphere-asthenosphere boundary beneath Axial Volcano

Corresponding Author: Professor Graham Kent

Version 0:

Reviewer comments:

Referee #1

(Remarks to the Author)
Review by William Chadwick

General comments on the manuscript:

This paper presents the results of a 3D multichannel seismic survey of Axial Seamount, the most active volcano in the NE Pacific ocean and probably the best monitored and submarine volcano on the planet. This is the first publication of the 3D seismic dataset, except for meeting abstracts. This is a fascinating and provocative paper. I would even describe it as “transformative”. It is one of the most exciting and interesting that I’ve read in a long time, and I enthusiastically endorse it as worthy of publication in Nature. I recommend publication after minor to moderate revision, after the authors consider the comments below and the suggested edits in the annotated manuscript file.

In general, the paper is very well-written and was a pleasure to read. The figures are excellent with a few exceptions (described below). However, there are a few specific areas where the manuscript could be improved and I have some questions and comments that may challenge some of the interpretations the authors present.

My specific suggestions are below:

1) First, I think the abstract (the 1st paragraph) and the text needs to find a way to relate this study to previous ones using seismic reflection data at Axial Seamount (from the 2D survey in 2002). More specifically, it needs to relate the LAB (lithosphere-asthenosphere boundary) concept with the idea of a “magma reservoir” (which is not even mentioned in the abstract, but was a big focus in previous work). Is the idea of a “magma reservoir” even relevant to Axial in the face of these new results? I’m wondering if “magma reservoir” should be re-imagined or re-named to something like “magma domain”, as suggested by Sigmundsson (2016) to refer to a volume within the crust within which melt bodies are distributed in space (& time)? So, I’m thinking perhaps the last sentence of the abstract should be changed to one like this: “In this context, the LAB can be viewed as the upper surface of a “magma domain”, a volume within which melt bodies reside (replacing the concept of a single “magma reservoir”).”

Reference:

Sigmundsson, F. (2016), New insights into magma plumbing along rift systems from detailed observations of eruptive behavior at Axial volcano, *Geophys. Res. Lett.*, 43, doi:10.1002/2016GL071884.

2) Another interpretation in the manuscript that I’m wondering about is whether Funnel A & B could be interpreted as being just one BIG Funnel with a saddle in the middle? See related comments in the annotated manuscript at the top of page 3. Along these lines, I’m also wondering if the “saddle” in the AML/LAB under the caldera could be interpreted as depressed due to repeated caldera collapse from above? If so, this would be another reason to consider the Funnel A & B structure as one surface instead of two. There isn’t really a break between Funnels A & B, is there? Why divide them into two? Don’t be married to the idea that your “Funnels” have to look like funnels!

3) Then there are two sections in the text that really stand out as weak and speculative, and I think they should both be omitted, because they detract from the rest of the paper. The first is the paragraph at the bottom of page 4. I think the analogy to Kilauea is a poor one. The south flank failure at Kilauea is due to rift zone spreading and a decollement-like fault at the base of the volcano sliding over thick sediments on a very old plate. There are huge normal faults on Kilauea’s south flank, occasional major earthquakes ($M > 6$), and LOTS of microseismicity that provide ample evidence of these processes. Nothing

like that is evident at Axial. I would omit this entire paragraph, as it is unconvincing and takes away from the rest of the paper. I would also omit the mentions of “failure surfaces” in the abstract and on the bottom of page 6.

The second weak and speculative section in the text that I would omit is on the bottom of page 5 and the top of page 6 (highlighted in the annotated manuscript). There is no way to know how the seismic reflection amplitudes might have changed with time at Axial, unless you compare the 2002 2D and 2019 3D surveys, so to speculate about it seems pointless. It is also unknown how or if the shallow distribution of melt in the subsurface changes between eruptions that intrude down the south vs. north rift zones. It seems equally likely to me that the source zone is the same for both, and only the direction of dike propagation differs.

Instead, I would replace these weak parts of the manuscript with a discussion of two subjects that seem very important but are missing from the current text:

a) The first would be a clear and concise discussion of how these new results compare to previous specific results published based on the 2D seismic survey in 2002. This is related to my comment #1 above (but here referring to adding to the text instead of the abstract). For example, Yang et al. (2024) image a magma reservoir beneath the SW western wall of the caldera, but none is evident in that location in the 3D survey. Does the 3D survey indicate it is not real? What about the other reservoirs imaged by Yang et al. (2024) and their interpreted connections? Likewise, the Arnulf et al. (2014; 2018) papers describe a Main Magma Reservoir (MMR) beneath the caldera, and a Secondary Magma Reservoir (SMR) located about 5 km to the southeast. Clearly, the MMR is the same as the combined “Funnel A + B” structures in this paper – but I would like to see that authors say that explicitly in the manuscript text. But what about the SMR? Is it real? Was it just part of Funnel B but misidentified as a separate reservoir? (it is not located far enough east to be Funnel C, as far as I can tell) I’d like to see a definitive statement of what the (better) 3D survey supports or does not support from the (worse) 2D survey results. This strikes me as important for the research community to know, going forward, because it has real-world consequences for where people will focus future research efforts.

b) The second subject that is missing and deserves some discussion in the manuscript is briefly comparing the 3D results to results from geodetic and seismic monitoring. There are some interesting comparisons to be made to the distribution and density of seismicity at Axial (see Wilcock et al., 2016; Waldhauser et al., 2020), particularly because the eastern edge of the caldera is where most earthquakes occur (near the “melt ribbon” in this paper). Also, a paper by Wang et al. (in revision at GRL, but previous results are in AGU abstracts) describes “mixed frequency earthquakes” from this same area that were detected before the 2015 eruption, and interpreted to be associated with the intrusion of the dike that fed the eruption. This has obvious connections to the “initiation zone” discussed in the text (see related comment in the annotated manuscript in the middle of page 3).

In regards to the geodetic results, there is a long-standing but interesting apparent mis-match between the location of maximum vertical deformation, which is at the center of the caldera, and the location(s) of highest melt concentration interpreted from seismic reflection surveys, which is along the eastern edge of the caldera and distinctly SE of the caldera (first from the 2D survey, and now in the 3D survey as well). Both observations are very repeatable and robust, but seemingly at odds. It is still a bit unclear to me how these two sets of observations might be reconciled with our current understanding of the Axial magmatic system, but I would be interested to hear if the authors have any new ideas. To me, it is not important that the authors have an “answer”; it would be fine to say “we don’t understand the reasons for this discrepancy”; I think it’s more important to acknowledge the issue, again pointing the way to future research problems that still need to be resolved. The recent paper by Sleed et al. (2024) in JGR would be good to reference related to this issue. (see related comment in the annotated manuscript at the top of page 4)

References:

Wilcock, W. S. D., M. Tolstoy, F. Waldhauser, C. Garcia, Y. J. Tan, D. R. Bohnenstiehl, J. Caplan-Auerbach, R. P. Dziak, A. F. Arnulf, and M. E. Mann (2016), Seismic constraints on caldera dynamics from the 2015 Axial Seamount eruption, *Science*, 354(6318), 1395-1399, doi:10.1126/science.aah5563.

Waldhauser, F., W. S. D. Wilcock, M. Tolstoy, C. Baillard, Y. J. Tan, and D. P. Schaff (2020), Precision seismic monitoring and analysis at Axial Seamount using a real-time double-difference system, *Journal of Geophysical Research: Solid Earth*, 125, e2019JB018796, doi:10.1029/2019JB018796.

Wang, K., F. Waldhauser, M. Tolstoy, D. P. Schaff, T. Sawi, W. S. D. Wilcock, and Y. J. Tan (in revision), Volcanic precursor revealed by machine learning offers new eruption forecasting capability, *Geophys. Res. Lett.*, doi: 10.22541/essoar.170758176.65197692/v1.

Sleed, S., M. Wei, S. L. Nooner, W. W. Chadwick Jr., D. W. Caress, and J. W. Beeson (2024), Compartmentalization of Axial Seamount’s magma reservoir inferred by analytical and numerical deformation modeling with realistic geometry, *Journal of Geophysical Research: Solid Earth*, 129(5), e2023JB028414, doi:10.1029/2023JB028414.

Comments on the Figures:

Figure 1 – The vent field name labels should be distinguished from - and less prominent than - the Funnel labels. Maybe try them without a white box behind them or perhaps a different font size and color; adding arrows between the names and the vent field symbols would also be good, since their connections are not clear as is. Actually arrows between the Funnel labels and what they refer to would also be good to add. The label for the CoAxial segment should be rotated to be parallel with the segment axis. Consider adding the Vance segment axis with a similar label.

Figure 2 – The locations of the two profiles shown in this figure should be indicated on Figure 1. The text labels would be more effective if arrows were added for each of them to point to the exact features in the profiles. Try the labels without the white box behind them.

Figure 3 – The color scale lacks values and units (it needs at least “high” and “low” to be added), and needs to be wider and more prominent. The same is true for all the other figures that include it.

Figure 4 – The exact locations of the hydrothermal vent fields are difficult to see in this figure. I'd suggest using arrows or perhaps a more prominent symbol that could be overlain on the voxels (like a white dot?).

Supplemental Figures 4 & 5 - I find it nearly impossible to understand the perspective of these 3D views without more information. Is it oblique? Or is it looking horizontally into the subsurface? (what angle from the horizontal is it?). I can't tell. I'm wondering if these views should be paired with another view showing bathymetry (possibly with seismic data in cross-section) from exactly the same viewpoint, to help orient the viewer. Or would adding the outline of the caldera help? I think it would help orient the view geographically, but I can't tell because I don't understand the perspective of the figures.

Referee #2

(Remarks to the Author)

The authors describe and interpret a 40 x 16.3 km² 3D seismic reflection survey taken across the Axial Volcano on the Juan de Fuca Ridge, separating the Pacific and Juan de Fuca plates offshore the Central Cascades volcanic arc, USA. The investigation identifies Axial Melt Lenses and the subjacent Lithosphere-Asthenosphere boundary under the ridge system in 4 different locations.

Axial melt lenses have been identified in seismic reflection data previously beneath the JdF and the East Pacific Rise midocean ridges, however, deeper structures have been rather poorly imaged or imaged in 2D. Seismic reflection and tomography data suggest that a layer of velocity intermediate between the AML and lower oceanic crust underlies the AML in ridge segments, interpreted as a zone consisting of mush rich sills. The interpretation is that these low velocity mush layer sills freeze to form oceanic crustal layer 3, and contribute melt to the AML.

This paper identifies the lithosphere-asthenosphere boundary beneath and around the AML in the reflection data as well as zones of sills being injected below the AML which will eventually cool to form the bulk of the oceanic crust (layer 3). The authors describe the morphology of the magmatic systems beneath the AMLs as funnels which are inverted beneath the AML. They identify 4 funnels which in one case may feed melt from one to the other. The funnels in aggregate are large, extending ~20km along ridge, and 4-5km cross-ridge, and 5-6 km below the seafloor. They interpret the sides of the funnels as melt assimilation zones, where pre-existing oceanic crust is re-melted and reincorporated into the melt system. They illustrate the complex geometry using perspective views from the 3D data volume, and movies that page through the data, as well as with reflection strength images.

One very interesting aspect of their observations is, if one takes the AML as the top of the oceanic convection system (i.e. the LAB), then the seismic Moho lies deeper than the LAB, at the base of the sill system. I don't believe that this has been observed and interpreted elsewhere before, and is in part the result of the ambiguities arising in interpreting geodynamic systems based on style of heat flow, using seismic definitions based on velocity gradient changes. The authors' interpretation conforms to the long held geodynamic view of the oceanic LAB at the midocean ridge extending to (or almost to) the ocean floor. I almost think that they should choose new terminology because this is in the conceptually gray zone where the lithosphere and asthenosphere meet. Nonetheless like other aspects of this paper, this is a first rate observation.

Overall I found the paper to be well written and clear, the methodology sound, and the interpretations reasonable based on previous seismic work and other geophysical information. The work builds on decades of research along these lines at the JdF Ridge and several at or near to this site. The images are spectacular. The results are new, heretofore unseen but inferred structure of the mid-ocean ridges. I believe that this paper merits publication in Nature.

Version 1:

Reviewer comments:

Referee #1

(Remarks to the Author)

Review by William Chadwick

General comments on the revised manuscript:

In general, I think the authors have done a great job in their revisions and have addressed all the issues that I raised in my first review. I think the paper is now in excellent shape and will make a strong contribution. I only have relatively minor comments, suggestions below, with one exception: the map in the revised Figure 4a has the east and west directions mistakenly swapped, so this obviously needs to be corrected (and Figure 4c, needs to be re-oriented accordingly). I recommend the issues below be addressed in another round of minor revision before publication. I am including an

annotated manuscript with my review (with the figures removed to make the file size smaller), but it only contains the suggestions and comments listed below).

Specific comments keyed to line numbers in the revised manuscript:

Line 59: Omit “beneath Axial volcano”, because “Axial volcano” is repeated at the end of the sentence, and you don’t need both.

Line 82: Add “the” before “Magma Domain” in the sub-section title

Lines 95-96: Change “funnels shapes” to “funnel shapes”, and change “and does not” to “that do not”.

Line 109: Add “of the volcano summit” after “bathymetric plateau”

Lines 120-121: Add “, but there are no known recently erupted lavas above them.” at the end of the sentence ending with reference #18. I think this is worth noting since obviously funnels A+B erupt at the surface, but funnels C+D do not (yet), as far as we know.

Line 144: Change one sentence with semicolon into two sentences (“...strength⁴³. Other...”)

Line 158: Change “an” to “at”

Line 161: add (“a” before “wave”)

Line 168: Change “followed by” to “and”, and change “which may match” to “consistent with”

Line 187: Change “that of” to “that they represent”

Line 188: Change one sentence with semicolon into two sentences (“...(Fig. 5). This...”)

Lines 190-191: Change “summit volcano” to “volcano summit”

Line 196: Insert “(Fig. 3)” at end of sentence (“...Funnel C (Fig. 3)”)

Line 231: Change “that mimic” to “which mimics”

Line 233: Change one sentence with semicolon into two sentences (“...18-19⁵¹. Melt...”)

Line 235: Add “and freezes” after “migrates”

Line 253: Change “lavas flows” to “lava flows”

Line 266: Add “seafloor” before “topography”

Line 274: Change “scope” to “extent”

Line 294: Change “the central volcano” to “a central volcano”

Line 295: Change “such as the” to “such as at the”, and “Lucky Strike volcano” to “Lucky Strike segment”

Line 296: Change “50°28'E volcano” to “50°28'E segment”

Lines 298-299: Change “at other volcanic centers” to “in other mid-ocean ridge settings”

Line 310 (Figure 1 caption): Add “previously identified” before “Main Magma Reservoir”

Lines 317-318 (Figure 2 caption): change “this Inline” to “the Inline”, and “this Xline” to “the Xline”

Line 332: Omit “abruptly”

Figure 4: Note that east and west are mistakenly flipped on this map (Fig. 4a)! This needs to be corrected! Comparing this with the original Figure 4 (which was correct), it looks like the X-line axis needs to be reversed, or the map needs to be flipped top-to-bottom. Note that the seismic amplitude figures may have to be re-oriented accordingly. Panels of Figure 4 need “a”, “b”, and “c” labels. Omit “upper left”, etc to refer to each panel, because it can be confused with trying to point something out within the panel. In (a), I would add labels by each bracket saying “view angle of Fig. 4b” and “view angle of Fig. 4c”. Move the “enhanced melt supply” label away from the bracket to clarify it is not related. Also add a north arrow. Figure 4c is flip-flopped (and rotated) in its orientation to match the flip-flopped map in Fig. 4a. It needs to be placed beneath Figure 4b, and presented in landscape orientation, like 4b and like it was in the original manuscript as supplemental figure 5.

Line 343 (Figure 4 caption): Omit “(upper-left)”

Line 345 (Figure 4 caption): Change “3-D view” to “perspective view”

Line 346 (Figure 4 caption): Omit “(lower-left)”

Line 348 (Figure 4 caption): Change “a, orange arrow” to “orange arrow in (a)”

Line 349 (Figure 4 caption): Omit “(right)”

Line 351 (Figure 4 caption): Omit “(a)” in the middle of the line and replace it with “in (a)” at the end of the sentence.

Lines 357-359 (Figure 4 caption): This information only applies to Fig. 4a

Line 360 (Figure 4 caption): Add “3-D” before “tour”.

Line 379 (Extended Data Figure 1 caption): Change “apparent displacement” to “the offset”

Line 381 (Extended Data Figure 1 caption): Omit “abruptly”

Line 382 (Extended Data Figure 1 caption): Change “(~5km)” to “by ~5km”

Dear Editor

We very much appreciate the insight from Bill Chadwick, an anonymous reviewer and your comments to produce a more clear and focused manuscript, through addressing their concerns, ideas, and suggestions, etc... these will be addressed below.

Sincerely,

Dr. Graham Kent

Summary of Reviewer #1 main comments addressed:

- (1) We appreciated the suggestion regarding the use “magma domain” regarding the larger volume that hosts a variety magma bodies. The abstract has been modified to reflect this concept. (*Lines 24-26*)
- (2) We have included comments in the manuscript, highlighting the relationships between Funnels A and B and reflectivity beneath the caldera. It’s less likely that the deeper reflectivity represents a collapse, or deepening of this LAB boundary, but rather a host of smaller melt pockets “link” across the caldera at a similar depth as the shallowest parts of Funnels A & B. (*Lines 106-108*)
- (3) We have removed the two speculative sections as suggested by Reviewer #1. Both research topics are currently being investigated, but we agree that they detract from the main thrust of this manuscript.
- (4) We have addressed the relationship of previously recognized magma reservoirs MMR and SMR in relationship to the new framework of Funnels A, B, C and D. We also have discussed the apparent mismatch between geodetic measurements in the caldera relative to melt availability maps; this observation may be potentially related to “weak” ring faults that encircle the caldera. (*Lines 104-106, Lines 338-339, MMR & SMR; Lines 174-179, ring faults*)

- (5) We have mentioned that we do not see any of the large-scale magmatic bodies (in our imagery) that were reported by Yang et al.; if these bodies exist, they would be very small relative to what has been imaged by this experiment. (*Lines 120-122*)
- (6) We have also included recent seismicity results and their relationship to the “melt ribbon” along the southeast caldera wall (Wilcock et al.; Waldhauser et al.; Wang et al.). (*Lines 135-139*)

Figure modifications:

Figure 1. We have modified labeling of both hydrothermal fields and Funnels A-D per suggestions of Reviewer #1. We have now labeled North/South Rift Zones and CoAxial and oriented both (CoAxial and NRZ/ SRZ) labels to be ridge parallel. We also locate profiles #286 and #2424 (as shown in Figure 2a and b) on the main map displayed in Figure 1. Lastly, we have placed outlines of the MMR and SMR (white dashed boundary) on the larger map.

Figure 2. As previously noted, these profile locations are now shown in Figure 1. Arrows now highlight sill truncation geometries. We failed to find a suitable letter color for LAB, etc... that works without the white box underlay. To minimize more arrows, we now rotate text boxes to provide a better association with the highlighted features.

Figure 3. Color bar has been widened and added to similar figures; color bar is now spatially associated with the time-axis just left of the bar.

Figure 4. Original Fig. 4 and Supp. Figs 4 and 5 have been combined into a new Figure 4 to help provide better geometrical orientation of voxel-based “melt availability” maps (new Figs 4a and 4b). Arrow (colored) and image bracket (Fig. 4a) are now associated with color frame of Figs. 4b and 4c. Vent fields (Fig. 4a) have been placed as the top layer (still magenta circles) and now are more prominent per Reviewer#1 request. We have also redrawn the 3-D compass avatars (Figs. 4b and 4c) to provide a clearer sense of orientation.

Reviewer #1 Copy Edit:

Otherwise, the authors were super appreciative of Reviewer #1 copy edits and nearly all were adopted outright (see provided comparison document between submitted and revised manuscript).

Summary of Reviewer # 2 main comment’s addressed:

Reviewer #2 would appreciate how much we went back ‘n forth on terminology of this newly imaged interface. Some colleagues appreciated the “crustal LAB” terminology, while a few were at best hoping we used something else. One of the observations we are hoping to make in the future is getting high quality seismic imagery of the Moho beneath the crustal LAB; at present, such an interface is stuck within the seafloor reflection multiple, so we are currently trying to figure out a workable demultiple scheme to get a decent image of the Moho beneath Axial volcano. This constraint would help answer whether the Moho is fully formed beneath the “crustal LAB”, or maybe it is only fully “set in” after the crust-mantle interface moves away from the ridge axis and hence outside the LAB boundary. We agree that this would be a very fundamental observation. For now, we are using this 3-D dataset to peel back layers of crustal the “onion”, and perhaps overtime someone thinks of a better descriptor (if needed). We do, however, like Reviewer #1’s suggestion of the use of “magma domain” instead of magma chamber for the envelope that bounds and contains all magmatic bodies therein. Maybe we are all converging....

Editor Remarks

Length: The present text length of your manuscript is somewhat longer than our usual length limits for a short Nature Article. We do have some flexibility and could allow you to maintain the present main text length, however we will need to consider this a firm upper limit (although you have ample room to expand your online Methods section – see below for details). Please also include line numbering when submitting your revised manuscript and upload your movies to our Manuscript Tracking System.

We have kept the word length (3722 revised vs. 3793 original) of the “core” article a bit shorter relative to the original manuscript per your request.

Title: Your present title is somewhat longer than we can accommodate. Would simply “Melt focusing along the lithosphere-asthenosphere boundary at Axial Volcano” suffice? Please feel free to suggest an alternative title, bearing in mind that it should be 75 characters or less in length (including spaces) and not contain punctuation.

Thanks for the suggestion; this title captures the essence of the paper.

Sub-headings: In our new all-Article format, we encourage several sub-headings within the main text of short Articles to break up the text. Such sub-headings may be up to 40 characters in length, including spaces, but please avoid generic sub-headings such as ‘Discussion’ or ‘Conclusions’.

Per your suggestion, we have added 3 new sub-headings: “3-D Structure of Magma Domain”, “Melt Focusing and Assimilation” and “Temporal Variability of the LAB”

Methods: At the end of the main text document (after the main figure legends), there should be a section entitled “Methods”, which provides a more detailed discussion of the additional methodological information that would allow other researchers to replicate the results (we define “Methods” quite broadly, so this is not limited to details of experimental protocols – supplementary discussion and analysis can also be included). The Methods section will not appear in the print version but will be fully copy-edited and appear online in the full-text HTML and PDF versions. The Methods section should be written as concisely as possible but should contain all elements necessary to allow interpretation and reproduction of the results. If there are additional references in the Methods section, their numbering should continue from the last reference in the main paper, and the list should follow the Methods section.

We have moved the Methods section at the end of main text (after the figure captions) as suggested by the Editor. The Methods section does contain all the detailed information so that the readers could replicate our results.

Main Text Statements: We require authors to provide a detailed Author Contribution statement immediately after the acknowledgements; the specific contributions of each author must be listed. It is also a condition of publication that authors include an Author Information statement indicating how to access information regarding reprints and permissions, stating whether or not there is a financial or non-financial competing interest, and naming the author to whom correspondence and requests for materials should be addressed. Please ensure that this section is included in the manuscript file after the Methods (but before the Extended Data legends) - it will not appear in the print version but will appear online in the full-text HTML and PDF versions. For details of “end note” style and an example see <https://www.nature.com/nature/for-authors/formatting-guide>.

We have included Author Contribution statement immediately after the Acknowledgements.

Display items: We ask that you take stock of all the data that have been generated throughout the review process and ensure that only the data most central to the conclusions are presented in the main figures. Figures should be comprehensible to readers in other or related disciplines, and assist their understanding of the paper. We encourage authors who are describing complex processes to include a schematic of the main finding as part of the Extended Data to aid readers unfamiliar with the immediate discipline. Figures should be as small and simple as is compatible with clarity. All panels of a figure should be logically connected; each panel of a multipart figure should be sized so that the whole figure can be reduced by the same amount and reproduced on the printed page at the smallest size at which essential details are visible. For guidance, *Nature's* standard figure sizes are 89 mm (one column), 120 mm (one and a half columns), or exceptionally 183 mm (two columns) wide; the full depth of a *Nature* page is 247 mm. All panels of figures should be presented on a single page and assembled into a rectangular shape for publication; please indicate any essential alignments (parts horizontal, vertical, spacings of stereo pairs, etc.). Tables should be prepared using the Table menu in Microsoft Word.

We have now used Extended Data Figures 1-3 to replace Supplemental Figures 1-3.

Figure formatting: Lettering in all figures (labelling of axes and so on) should be in uniform, sans-serif font, in lower-case type, and large enough to permit substantial reduction for publication (minimum font size 5 pt). Separate parts of a figure are labelled a, b, etc. Units have a single space between the number and the unit, and follow SI nomenclature or the nomenclature common to a particular field. Thousands are separated by commas (1,000). Unusual units or abbreviations are defined in the legend. Scale bars rather than magnification factors should be used.

Done.

ORCID: *Nature* is committed to improving transparency in authorship. As part of our efforts in this direction, we are now requesting that all authors identified as ‘corresponding author’ create and link their Open Researcher and Contributor Identifier (ORCID) with their account on the Manuscript Tracking System prior to acceptance. ORCID helps the scientific community achieve unambiguous attribution of all scholarly contributions. You can create and link your ORCID from the home page of the Manuscript Tracking System by clicking on ‘Modify my Springer Nature account’ and following the instructions in the link below. If you experience problems in linking your ORCID, please contact the Platform Support Helpdesk.

Done.

I have also signed up and linked my ORCID account to my Springer Nature account.

* Include a point-by-point response to the issues raised by the referees.

The point by point response to the reviewers are included below

* Ensure it complies with our format requirements as set out at <<https://www.nature.com/nature/for-authors/formatting-guide>>.

We have read the format requirements and the paper complies with them.

Referee #1:

Review by William Chadwick

General comments on the manuscript:

This paper presents the results of a 3D multichannel seismic survey of Axial Seamount, the most active volcano in the NE Pacific ocean and probably the best monitored and submarine volcano on the planet. This is the first publication of the 3D seismic dataset, except for meeting abstracts. This is a fascinating and provocative paper. I would even describe it as “transformative”. It is one of the most exciting and interesting that I’ve read in a long time, and I enthusiastically endorse it as worthy of publication in Nature. I recommend publication after minor to moderate revision, after the authors consider the comments below and the suggested edits in the annotated manuscript file.

In general, the paper is very well-written and was a pleasure to read. The figures are excellent with a few exceptions (described below). However, there are a few specific areas where the manuscript could be improved and I have some questions and comments that may challenge some of the interpretations the authors present.

We thank the reviewer for frank positive and constructive remarks.

My specific suggestions are below:

1) First, I think the abstract (the 1st paragraph) and the text needs to find a way to relate this study to previous ones using seismic reflection data at Axial Seamount (from the 2D survey in 2002). More specifically, it needs to relate the LAB (lithosphere-asthenosphere boundary) concept with the idea of a “magma reservoir” (which is not even mentioned in the abstract, but was a big focus in previous work). Is the idea of a “magma reservoir” even relevant to Axial in the face of these new results? I’m wondering if “magma reservoir” should be re-imagined or re-named to something like “magma domain”, as suggested by Sigmundsson (2016) to refer to a volume within the crust within which melt bodies are distributed in space (& time)? So, I’m thinking perhaps the last sentence of the abstract should be changed to one like this: “In this context, the LAB can be viewed as the upper surface of a “magma domain”, a volume within which melt bodies reside (replacing the concept of a single “magma reservoir”).”

Thanks to the reviewer for this intriguing remark about ‘magma reservoir’ versus ‘magma domain’. Indeed, as the LAB is a thermally-controlled boundary as it encompasses the magma reservoir, even though there might be some sills intruded in the crust away from this main melt body and hence we have accepted the reviewers terminology of ‘magma domain’ and have rephased the text and included in the figures.

Reference:

Sigmundsson, F. (2016), New insights into magma plumbing along rift systems from detailed observations of eruptive behavior at Axial volcano, Geophys. Res. Lett., 43, doi:10.1002/2016GL071884.

We have incorporated the idea from this paper and have cited this paper.

2) Another interpretation in the manuscript that I’m wondering about is whether Funnel A & B could be interpreted as being just one BIG Funnel with a saddle in the middle? See related comments in the annotated manuscript at the top of page 3. Along these lines, I’m also wondering if the “saddle” in the AML/LAB under the caldera could be interpreted as depressed due to repeated caldera collapse from above? If so, this would be another reason to consider the Funnel A & B structure as one surface instead of two. There isn’t really a break between Funnels A & B, is there? Why divide them into two? Don’t be married to the idea that your “Funnels” have to look like funnels!

We have included comments in the manuscript, highlighting the relationships between Funnels A and B and reflectivity beneath the caldera. It's less likely that the deeper reflectivity represents a collapse, or deepening of this LAB boundary, but rather a host of smaller melt pockets "link" across the caldera at a similar depth as the shallowest parts of Funnels A and B.

3) Then there are two sections in the text that really stand out as weak and speculative, and I think they should both be omitted, because they detract from the rest of the paper. The first is the paragraph at the bottom of page 4. I think the analogy to Kilauea is a poor one. The south flank failure at Kilauea is due to rift zone spreading and a decollement-like fault at the base of the volcano sliding over thick sediments on a very old plate. There are huge normal faults on Kilauea's south flank, occasional major earthquakes ($M > 6$), and LOTS of microseismicity that provide ample evidence of these processes. Nothing like that is evident at Axial. I would omit this entire paragraph, as it is unconvincing and takes away from the rest of the paper. I would also omit the mentions of "failure surfaces" in the abstract and on the bottom of page 6.

The second weak and speculative section in the text that I would omit is on the bottom of page 5 and the top of page 6 (highlighted in the annotated manuscript). There is no way to know how the seismic reflection amplitudes might have changed with time at Axial, unless you compare the 2002 2D and 2019 3D surveys, so to speculate about it seems pointless. It is also unknown how or if the shallow distribution of melt in the subsurface changes between eruptions that intrude down the south vs. north rift zones. It seems equally likely to me that the source zone is the same for both, and only the direction of dike propagation differs.

We agree with the reviewer and have removed the above two speculative sections as suggested.

Instead, I would replace these weak parts of the manuscript with a discussion of two subjects that seem very important but are missing from the current text:

a) The first would be a clear and concise discussion of how these new results compare to previous specific results published based on the 2D seismic survey in 2002. This is related to my comment #1 above (but here referring to adding to the text instead of the abstract). For example, Yang et al. (2024) image a magma reservoir beneath the SW western wall of the caldera, but none is evident in that location in the 3D survey. Does the 3D survey indicate it is not real? What about the other reservoirs imaged by Yang et al. (2024) and their interpreted connections? Likewise, the Arnulf et al. (2014; 2018) papers describe a Main Magma Reservoir (MMR) beneath the caldera, and a Secondary Magma Reservoir (SMR) located about 5 km to the southeast. Clearly, the MMR is the same as the combined "Funnel A + B" structures in this paper – but I would like to see that authors say that explicitly in the manuscript text. But what about the SMR? Is it real? Was it just part of Funnel B but misidentified as a separate reservoir? (it is not located far enough east to be Funnel C, as far as I can tell) I'd like to see a definitive statement of what the (better) 3D survey supports or does not support from the (worse) 2D survey results. This strikes me as important for the research community to know, going forward, because it has real-world consequences for where people will focus future research efforts.

We have included the discussion on the relationship of previously recognized magma reservoirs MMR and SMR (Arnulf et al., 2014; 2018) in relationship to the new framework of Funnels A, B, C and D. Outline of MMR and SMR are now outlined on Figure 1. We have now mentioned that we do not see any of the large-scale magmatic bodies (in our post-stack, time-migrated image) that were reported by Yang et al.; if these bodies exist, they would be very small relative to what has been imaged by this experiment.

b) The second subject that is missing and deserves some discussion in the manuscript is briefly comparing

the 3D results to results from geodetic and seismic monitoring. There are some interesting comparisons to be made to the distribution and density of seismicity at Axial (see Wilcock et al., 2016; Waldhauser et al., 2020), particularly because the eastern edge of the caldera is where most earthquakes occur (near the “melt ribbon” in this paper). Also, a paper by Wang et al. (in revision at GRL, but previous results are in AGU abstracts) describes “mixed frequency earthquakes” from this same area that were detected before the 2015 eruption, and interpreted to be associated with the intrusion of the dike that fed the eruption. This has obvious connections to the “initiation zone” discussed in the text (see related comment in the annotated manuscript in the middle of page 3).

We have added these earthquake-related observations to the revised manuscript.

In regards to the geodetic results, there is a long-standing but interesting apparent mis-match between the location of maximum vertical deformation, which is at the center of the caldera, and the location(s) of highest melt concentration interpreted from seismic reflection surveys, which is along the eastern edge of the caldera and distinctly SE of the caldera (first from the 2D survey, and now in the 3D survey as well). Both observations are very repeatable and robust, but seemingly at odds. It is still a bit unclear to me how these two sets of observations might be reconciled with our current understanding of the Axial magmatic system, but I would be interested to hear if the authors have any new ideas. To me, it is not important that the authors have an “answer”; it would be fine to say “we don’t understand the reasons for this discrepancy”; I think it’s more important to acknowledge the issue, again pointing the way to future research problems that still need to be resolved. The recent paper by Sleat et al. (2024) in JGR would be good to reference related to this issue. (see related comment in the annotated manuscript at the top of page 4)

We also have discussed the apparent mismatch between geodetic measurements in the caldera relative to melt availability maps; this observation may be potentially related to “weak” ring faults that encircle the caldera. Current 3-D prestack depth migration (PSDM) of the data volume may help further develop ideas on this conundrum/mismatch with proper depth imaging of this system and association with seismicity at depth.

References:

Wilcock, W. S. D., M. Tolstoy, F. Waldhauser, C. Garcia, Y. J. Tan, D. R. Bohnenstiehl, J. Caplan-Auerbach, R. P. Dziak, A. F. Arnulf, and M. E. Mann (2016), Seismic constraints on caldera dynamics from the 2015 Axial Seamount eruption, *Science*, 354(6318), 1395-1399, doi:10.1126/science.aah5563.

Waldhauser, F., W. S. D. Wilcock, M. Tolstoy, C. Baillard, Y. J. Tan, and D. P. Schaff (2020), Precision seismic monitoring and analysis at Axial Seamount using a real-time double-difference system, *Journal of Geophysical Research: Solid Earth*, 125, e2019JB018796, doi:10.1029/2019JB018796.

Wang, K., F. Waldhauser, M. Tolstoy, D. P. Schaff, T. Sawi, W. S. D. Wilcock, and Y. J. Tan (in revision), Volcanic precursor revealed by machine learning offers new eruption forecasting capability, *Geophys. Res. Lett.*, doi: 10.22541/essoar.170758176.65197692/v1.

Sleat, S., M. Wei, S. L. Nooner, W. W. Chadwick Jr., D. W. Caress, and J. W. Beeson (2024), Compartmentalization of Axial Seamount’s magma reservoir inferred by analytical and numerical deformation modeling with realistic geometry, *Journal of Geophysical Research: Solid Earth*, 129(5), e2023JB028414, doi:10.1029/2023JB028414.

We have included these references in the revised version.

Comments on the Figures:

Figure 1 – The vent field name labels should be distinguished from - and less prominent than - the Funnel labels. Maybe try them without a white box behind them or perhaps a different font size and color; adding arrows between the names and the vent field symbols would also be good, since their connections are not clear as is. Actually arrows between the Funnel labels and what they refer to would also be good to add. The label for the CoAxial segment should be rotated to be parallel with the segment axis. Consider adding the Vance segment axis with a similar label.

We have modified labeling of both hydrothermal fields and Funnels per suggestions of Reviewer #1. We have now labeled the South Rift Zone (not Vance) and oriented both (CoAxial and SRZ/NRZ) labels to be ridge parallel. MMR and SMR boundaries have been added to the larger map.

Figure 2 – The locations of the two profiles shown in this figure should be indicated on Figure 1. The text labels would be more effective if arrows were added for each of them to point to the exact features in the profiles. Try the labels without the white box behind them.

We have located profiles #286 and #2424 (as shown in Figure 2 a and b) on the large map displayed in Figure 1. Arrows now highlight sill truncation geometries. We failed to find a suitable letter color for LAB, etc... that works without the white box underlay. To minimize more arrows, we now rotate text boxes that better provides association with the features we're trying to highlight.

Figure 3 – The color scale lacks values and units (it needs at least “high” and “low” to be added), and needs to be wider and more prominent. The same is true for all the other figures that include it.

Color bar has been widened and added to all the figures; color bar is now spatially associated with the time-axis just left of the bar.

Figure 4 – The exact locations of the hydrothermal vent fields are difficult to see in this figure. I'd suggest using arrows or perhaps a more prominent symbol that could be overlain on the voxels (like a white dot?).

Original Fig. 4 and Supp. Figs 4 and 5 have been combined into a new Figure 4 to help provide better geometrical orientation of voxel-based “melt availability” maps (new Figs 4b and 4c). Arrow (colored) and image bracket (Fig. 4a) are now associated with color frame of Figs 4b and 4c. Vent fields (Fig. 4a) have been placed as the top layer (still magenta circles) and now are more prominent per Reviewer#1 request. We have also redrawn the 3-D compass avatars (Figs. 4b and 4c) to provide a clearer sense of orientation.

Supplemental Figures 4 & 5 - I find it nearly impossible to understand the perspective of these 3D views without more information. Is it oblique? Or is it looking horizontally into the subsurface? (what angle from the horizontal is it?). I can't tell. I'm wondering if these views should be paired with another view showing bathymetry (possibly with seismic data in cross-section) from exactly the same viewpoint, to help orient the viewer. Or would adding the outline of the caldera help? I think it would help orient the view geographically, but I can't tell because I don't understand the perspective of the figures.

See new Figure 4.

Copy Edit: We are very appreciative of Reviewer #1 copy edits, which has been included in full in the revised version. Thank you.

Referee #2:

The authors describe and interpret a 40 x 16.3 km² 3D seismic reflection survey taken across the Axial Volcano on the Juan de Fuca Ridge, separating the Pacific and Juan de Fuca plates offshore the Central Cascades volcanic arc, USA. The investigation identifies Axial Melt Lenses and the subjacent Lithosphere-Asthenosphere boundary under the ridge system in 4 different locations.

Axial melt lenses have been identified in seismic reflection data previously beneath the JdF and the East Pacific Rise midocean ridges, however, deeper structures have been rather poorly imaged or imaged in 2D. Seismic reflection and tomography data suggest that a layer of velocity intermediate between the AML and lower oceanic crust underlies the AML in ridge segments, interpreted as a zone consisting of mush rich sills. The interpretation is that these low velocity mush layer sills freeze to form oceanic crustal layer 3, and contribute melt to the AML.

This paper identifies the lithosphere-asthenosphere boundary beneath and around the AML in the reflection data as well as zones of sills being injected below the AML which will eventually cool to form the bulk of the oceanic crust (layer 3). The authors describe the morphology of the magmatic systems beneath the AMLs as funnels which are inverted beneath the AML. They identify 4 funnels which in one case may feed melt from one to the other. The funnels in aggregate are large, extending ~20km along ridge, and 4-5km cross-ridge, and 5-6 km below the seafloor. They interpret the sides of the funnels as melt assimilation zones, where pre-existing oceanic crust is re-melted and reincorporated into the melt system. They illustrate the complex geometry using perspective views from the 3D data volume, and movies that page through the data, as well as with reflection strength images.

One very interesting aspect of their observations is, if one takes the AML as the top of the oceanic convection system (i.e. the LAB), then the seismic Moho lies deeper than the LAB, at the base of the sill system. I don't believe that this has been observed and interpreted elsewhere before, and is in part the result of the ambiguities arising in interpreting geodynamic systems based on style of heat flow, using seismic definitions based on velocity gradient changes. The authors interpretation conforms to the long held geodynamic view of the oceanic LAB at the midocean ridge extending to (or almost to) the ocean floor. I almost think that they should choose new terminology because this is in the conceptually gray zone where the lithosphere and asthenosphere meet. Nonetheless like other aspects of this paper, this is a first rate observation.

Overall I found the paper to be well written and clear, the methodology sound, and the interpretations reasonable based on previous seismic work and other geophysical information. The work builds on decades of research along these lines at the JdF Ridge and several at or near to this site. The images are spectacular. The results are new, heretofore unseen but inferred structure of the mid-ocean ridges. I believe that this paper merits publication in *Nature*.

We thank reviewer for positive comments. Indeed, the reflection image of the AML/LAB down to ~6 km depth is very exciting. It would be tempted to come up with a new terminology, but already have AML, AMC, Melt Triangle, and LAB. As more and more evidence suggest that LAB might be underlain by melt, we feel the crustal LAB would be most appropriate. How this crustal LAB transitions to the mantle LAB below the Moho is the focus of a recently funded European Research Council Advanced Grant Award (<https://cordis.europa.eu/project/id/101141564>).

Feb. 24th, 2025

Dear Editor,

With respect to Reviewer #1's main concern in his second round (see directly below), I have mirrored (fixed) the image presentation of the bathymetry as originally shown in Figure 4a, but this required that I move the bathymetric panel to Fig. 4b to make a better geographical layout (basically swapped Figs 4a. & 4b; 4b now has the corrected (non-mirrored) image per Referee #1 request). We have also used his requested verbiage for "view angle of Fig. 4a,c; added (a), (b), and (c) to panels (not using "upper-left", etc.

Figure 4: Note that east and west are mistakenly flipped on this map (Fig. 4a)! This needs to be corrected! Comparing this with the original Figure 4 (which was correct), it looks like the X-line axis needs to be reversed, or the map needs to be flipped top-to-bottom. Note that the seismic amplitude figures may have to be re-oriented accordingly. Panels of Figure 4 need "a", "b", and "c" labels. Omit "upper left", etc to refer to each panel, because it can be confused with trying to point something out within the panel. In (a), I would add labels by each bracket saying "view angle of Fig. 4b" and "view angle of Fig. 4c". Move the "enhanced melt supply" label away from the bracket to clarify it is not related. Also add a north arrow. Figure 4c is flip-flopped (and rotated) in its orientation to match the flip-flopped map in Fig. 4a. It needs to be placed beneath Figure 4b, and presented in landscape orientation, like 4b and like it was in the original manuscript as supplemental figure 5.

We have adopted the minor (but useful) edits from Reviewer #1 as outlined below.

Reviewer#1 specific comments keyed to line numbers in the revised manuscript:

Line 59: Omit "beneath Axial volcano", because "Axial volcano" is repeated at the end of the sentence, and you don't need both. **We adopted Reviewer #1 suggested edit.**

Line 82: Add "the" before "Magma Domain" in the sub-section title.. **We adopted Reviewer #1 suggested edit.**

Lines 95-96: Change "funnels shapes" to "funnel shapes", and change "and does not" to "that do not". **We adopted Reviewer #1 suggested edit.**

Line 109: Add "of the volcano summit" after "bathymetric plateau" **We adopted Reviewer #1 suggested edit.**

Lines 120-121: Add ", but there are no known recently erupted lavas above them." at the end of the sentence ending with reference #18. I think this is worth noting since obviously funnels A+B erupt at the surface, but funnels C+D do not (yet), as far as we know. **We added Reviewer #1 suggested edit.**

Line 144: Change one sentence with semicolon into two sentences ("...strength⁴³. Other...") **We adopted Reviewer #1 suggested edit and broke into two sentences.**

Line 158: Change “an” to “at” **We adopted Reviewer #1 suggested edit.**

Line 161: add (“a” before “wave”) **We adopted Reviewer #1 suggested edit.**

Line 168: Change “followed by” to “and”, and change “which may match” to “consistent with” **We adopted Reviewer #1 suggested edit.**

Line 187: Change “that of” to “that they represent” **We adopted Reviewer #1 suggested edit.**

Line 188: Change one sentence with semicolon into two sentences (“...(Fig. 5). This...”) **We adopted Reviewer #1 suggested edit and broke into two sentences.**

Lines 190-191: Change “summit volcano” to “volcano summit” **We adopted Reviewer #1 suggested edit.**

Line 196: Insert “(Fig. 3)” at end of sentence (“...Funnel C (Fig. 3)”) **We adopted Reviewer #1 suggested edit.**

Line 231: Change “that mimic” to “which mimics” **We adopted Reviewer #1 suggested edit.**

Line 233: Change one sentence with semicolon into two sentences (“...18-19[°]51. Melt...”) **We adopted Reviewer #1 suggested edit.**

Line 235: Add “and freezes” after “migrates” **We adopted Reviewer #1 suggested edit and broke into two sentences.**

Line 253: Change “lavas flows” to “lava flows” **We adopted Reviewer #1 suggested edit.**

Line 266: Add “seafloor” before “topography” **We adopted Reviewer #1 suggested edit.**

Line 274: Change “scope” to “extent” **We adopted Reviewer #1 suggested edit.**

Line 294: Change “the central volcano” to “a central volcano” **We adopted Reviewer #1 suggested edit.**

Line 295: Change “such as the” to “such as at the”, and “Lucky Strike volcano” to “Lucky Strike segment” **We adopted Reviewer #1 suggested edit.**

Line 296: Change “50°28'E volcano” to “50°28'E segment” **We adopted Reviewer #1 suggested edit.**

Lines 298-299: Change “at other volcanic centers” to “in other mid-ocean ridge settings” **We adopted Reviewer #1 suggested edit.**

Line 310 (Figure 1 caption): Add “previously identified” before “Main Magma Reservoir” **We adopted Reviewer #1 suggested edit.**

Lines 317-318 (Figure 2 caption): change “this Inline” to “the Inline”, and “this Xline” to “the Xline” **We adopted Reviewer #1 suggested edit.**

Line 332: Omit “abruptly” **We adopted Reviewer #1 suggested edit.**

Line 343 (Figure 4 caption): Omit “(upper-left)” **We adopted Reviewer #1 suggested edit.**

Line 345 (Figure 4 caption): Change “3-D view” to “perspective view” **We adopted Reviewer #1 suggested edit.**

Line 346 (Figure 4 caption): Omit “(lower-left)” **We adopted Reviewer #1 suggested edit.**

Line 348 (Figure 4 caption): Change “a, orange arrow” to “orange arrow in (a)” **We adopted Reviewer #1 suggested edit.**

Line 349 (Figure 4 caption): Omit “(right)” **We adopted Reviewer #1 suggested edit.**

Line 351 (Figure 4 caption): Omit “(a)” in the middle of the line and replace it with “in (a)” at the end of the sentence. **We adopted Reviewer #1 suggested edit.**

Lines 357-359 (Figure 4 caption): This information only applies to Fig. 4a (**now 4b**).

Line 360 (Figure 4 caption): Add "3-D" before "tour". **We adopted Reviewer #1 suggested edit.**

Line 379 (Extended Data Figure 1 caption): Change "apparent displacement" to "the offset" **We adopted Reviewer #1 suggested edit.**

Line 381 (Extended Data Figure 1 caption): Omit "abruptly" **We adopted Reviewer #1 suggested edit.**

Line 382 (Extended Data Figure 1 caption): Change "(~5km)" to "by ~5km" **We adopted Reviewer #1 suggested edit.**

*****END*****

Sincerely,

Dr. Graham Kent